# Thermo-reversible gelation of self-assembled conducting polymer colloids

Vidhika S. Damani [1], Xinran Xie[2], Rachel E. Daso[2], Khushboo Suman [3,4], Masoud Ghasemi [5], Weiran Xie [1], Ruiheng Wu [2], Yuhang Wu[1], Calvin L. Chao [6], Julian E. Alberto[3], Casey M. Lorch[7], Ai-Nin Yang[8], Dan My Nguyen[8], Tulaja Shrestha[8], Kayla Otero [8], Chun-Yuan Lo[8], Darrin J. Pochan [1], Enrique D. Gomez [5], Jonathan Rivnay [2,9] & Laure V. Kayser [1,8] ✉

Electrically conductive hydrogels based on conducting polymers have found increased use in bioelectronics due to their low moduli that mimic biological tissues, their ability to transport both ionic and electronic charges, and their ease of processing in various form factors via printing or injection. Current approaches towards conductive hydrogels, however, rely on covalent and therefore irreversible crosslinking mechanisms. Here, we report a thermo-responsive conducting polymer (TR-CP) that undergoes a fully reversible non-covalent crosslinking at 35 °C within less than a minute to form conductive hydrogels. The TR-CP is based on a block polyelectrolyte complex, that self-assembles into well-defined colloidal particles in water which undergo an isovolumetric sol-gel transition just below physiological temperature. The hydrogels have tunable mechanical properties in the 20 to 200 Pa range, are stable at various pH and salt conditions, self-healing, injectable, and biocompatible in vitro and in vivo. We demonstrate that the TR-CPs can be used to fabricate sensitive, conformal and reusable electrodes for surface electromyography. This thermo-responsive material provides exciting opportunities for stimuli-responsive and adaptive bioelectronics.

Conducting polymers such as polyaniline (PANI), polypyrrole (PPy), and poly(3,4-ethylene dioxythiophene):poly(styrene sulfonate) (PEDOT:PSS) are versatile materials across bioelectronic applications. They have gained significant attention for their relative softness compared with inorganic materials, ability to transport both electronic and ionic charges, and ease of fabrication via solution processing[1]. Particularly, their deployment as scaffolds and hydrogels has enabled integration with biological systems[2–5], through the formation of electrically-conductive materials with properties typically only achievable in insulating polymers. These conductive hydrogels can have mechanical properties mimicking soft tissues, injectability, or adhesiveness, which makes them useful in soft electrodes for electromyography[4,6], wearable electrical stimulators[7], cardiac monitoring[8], in vivo nerve stimulation[9,10], and neural recording[11–13].

The most common approaches to preparing these functional conductive hydrogels are (1) blending a conducting polymer with a

[1]Department of Materials Science and Engineering, University of Delaware, Newark, DE, USA. [2]Department of Biomedical Engineering, Northwestern University, Evanston, IL, USA. [3]Department of Chemical and Biomolecular Engineering, University of Delaware, Newark, DE, USA. [4]Department of Chemical Engineering, Indian Institute of Technology Madras, Chennai, India. [5]Department of Chemical Engineering and Department of Materials Science and Engineering, The Pennsylvania State University, University Park, PA, USA. [6]Department of Surgery, Northwestern University Feinberg School of Medicine, Chicago, IL, USA. [7]Department of Biomedical Engineering, University of Delaware, Newark, DE, USA. [8]Department of Chemistry and Biochemistry, University of Delaware, Newark, DE, USA. [9]Department of Materials Science and Engineering, Northwestern University, Evanston, IL, USA. ✉e-mail: lkayser@udel.edu

non-conductive crosslinked hydrogel with the properties of interest[14], (2) blending a conductive polymer with a hydrogel precursor then crosslinking the blend[15,16], or (3) oxidatively polymerizing the conducting polymer in the hydrogel or its precursor solution[17]. While effective, all these approaches rely on covalent crosslinkers, leading to the lack of reversibility in the gelation process. Recently, Pappa and coworkers reported an example of crosslinker-free gelation using a mixture of chitosan, gelatin, and PEDOT:PSS[18]. However, these blends are liquid at physiological temperature, which limits their applicability in in vivo and in vitro bioelectronics.

Here, we achieved a reversible sol-gel transition in conductive polymers just below physiological temperature (35 °C) through a fundamentally different approach that does not rely on covalent crosslinkers nor composite blends. The key molecular design was the synthesis of a thermo-responsive block copolymer of PSS with poly(N-isopropylacrylamide) (PSS-$b$-PNIPAM) that was then used as a matrix to prepare PEDOT:PSS-$b$-PNIPAM polyelectrolyte complexes (Fig. 1a). Due to the block copolymer structure, these intrinsically thermo-responsive conducting polymers (TR-CP) showed self-assembly of well-defined and stable colloidal particles in water, which rapidly (<1 min) and reversibly formed a conductive hydrogel network above 35 °C. The TR-CP gels have good conductivity (~10 mS cm$^{-1}$) and are cytocompatible in vitro and in vivo as prepared or in composites. We show that the TR-CP are self-healing and injectable and demonstrate their utility as reusable electrodes for surface electromyography (s-EMG). Due to their stimuli-responsive properties, tolerance to a range of pH and salt concentrations, and cytocompatibility, the TR-CP could be transformative for further bioelectronic applications such as adaptive, patternable, injectable, and thermally degradable electrodes, scaffolds, and actuators.

## Results

### Design, synthesis, and rheological properties of the TR-CP

To obtain TR-CP, we chose PEDOT:PSS as the conductive component because it is water-dispersible, oxygen stable, displays high conductivity in the solid state, and is readily used and studied in

bioelectronics[19]. For the thermo-responsive component, PNIPAM was chosen because of its lower critical solution temperature (LCST) transition close to body temperature (32–35 °C)[20]. To promote the phase separation of hydrophobic and hydrophilic domains and facilitate PNIPAM inter-chain interactions above the LCST to form gels, we synthesized (Fig. S1–S5) and characterized (Fig. S6–S7) block copolymers of PSS with PNIPAM with varying molecular weights and block ratio (Table 1) by reversible addition-fragmentation chain transfer (RAFT) polymerization (SI Section 2.1). These block copolymers were then used as templates to synthesize PEDOT:PSS-$b$-PNIPAM (Figs. S3 and 1a) by oxidative polymerization of EDOT in water.

The phase change of these TR-CPs upon heating was monitored visually and by variable-temperature rheology (Fig. S8). The latter was used to determine the optimal formulation (PSS-$b$-PNIPAM block copolymer) for the formation of a fully reversible gel that was stable at physiological temperature. We found that the best TR-CP formulation had a mass ratio of PSS:PNIPAM of 1:2.3 (PEDOT:PSS$_{96}$-$b$-PNIPAM$_{440}$, Table 1, Entry 4). It formed a gel at $T_{gel}$ = 35 °C (Fig. 1b, c) as estimated from the crossover point between the storage (G') and loss (G'') moduli. The sol-gel transition was found to be isovolumetric (Fig. S9a) and very rapid, with a gelation time of only 41 s at 37 °C (Fig. 1d and Supplementary Movie 1). The storage modulus reached a plateau of 20 Pa at 37 °C, indicating the formation of a stable gel (Fig. 1c). The gel was also stable over a wide range of frequencies at 37 °C from 0.1 rad s$^{-1}$ to 20 rad s$^{-1}$ (Fig. S9b). To establish the reversibility of the sol-gel transition, a subsequent cooling ramp was performed. We observed that on cooling below the LCST, the storage and loss moduli returned to their original values in the solution state, with a slight hysteresis (Fig. S9c). Such hysteresis is observed commonly in other non-conductive thermo-responsive gels and can be eliminated by equilibration[21,22]. We established that the TR-CP retained its reversible sol-gel transition under multiple thermal cycling events, as seen over ten heat-cool cycles in the rheometer (Figs. 1e and S9d). It was found that G'' of the gel (50 °C) remained unchanged throughout the experiment. The storage modulus of the liquid increased slightly from 0.2 Pa to 0.4 Pa, likely caused by dehydration at prolonged exposure to

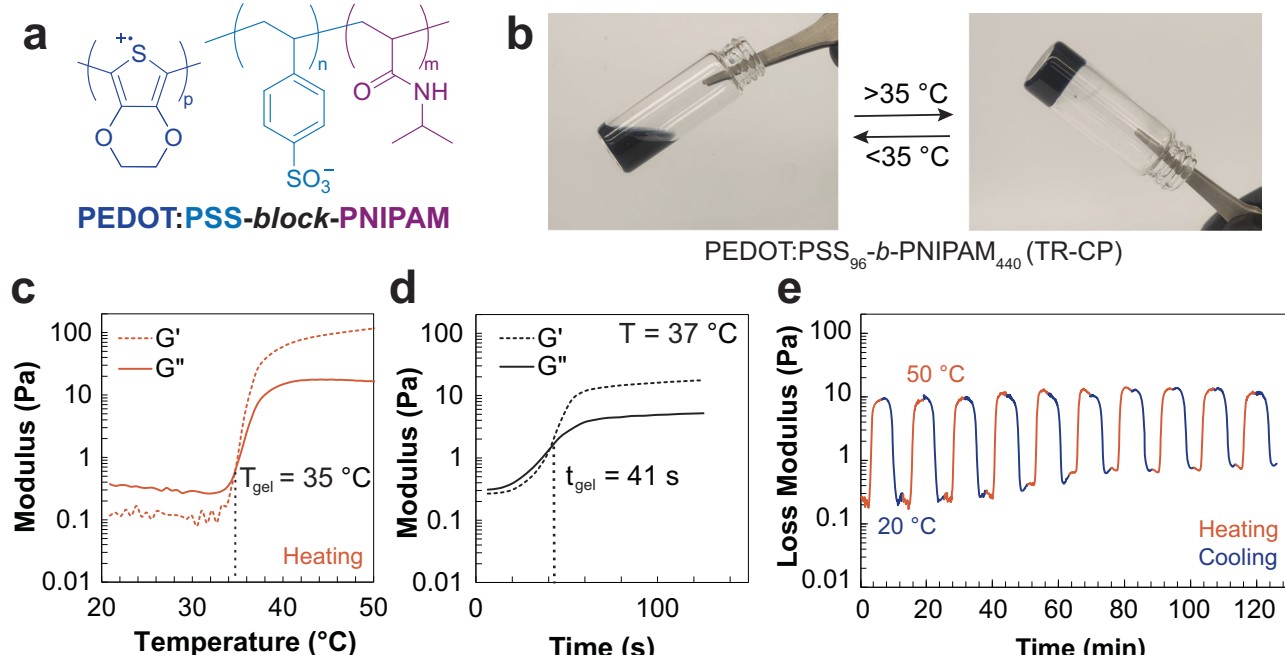

**Fig. 1 | Design and rheological properties of the thermo-responsive conductive polymers (TR-CP). a** Chemical structure of PEDOT:PSS-$b$-PNIPAM. **b** Picture of the sol-gel transition in the TR-CP PEDOT:PSS$_{96}$-$b$-PNIPAM$_{440}$ at 35 °C (3.6 wt% in water). **c** Variation in storage (G') and loss (G'') moduli upon heating from 20 °C to 50 °C (5 °C/min). **d** Change in storage (G') and loss (G'') moduli as a function of time, at a constant temperature of 37 °C. **e** Reversibility of the gelation and cycling stability over 10 heat-cool cycles from 20 °C to 50 °C on the rheometer.

**Table 1 | Synthesis of PSS-*b*-PNIPAM and thermo-responsive behavior of PEDOT:PSS-*b*-PNIPAM (TR-CP)**

| Entry | PSS-*b*-PNIPAM block copolymer | $M_n$ PSS (kg mol$^{-1}$) | | Đ[b] PSS | $M_n$ PSS-*b*-PNIPAM (kg mol$^{-1}$) | | Đ[b] PSS-*b*-PNIPAM | PSS:PNIPAM mass ratio[a] | Thermo-response of TR-CP at 37 °C [c] |
|---|---|---|---|---|---|---|---|---|---|
| | | Theo[a] | Meas[b] | | Theo[a] | Meas[b] | | | |
| 1 | PSS$_{95}$-*b*-PNIPAM$_{128}$ | 19.7 | 12.6 | 1.36 | 34.1 | 19.6 | 1.31 | 1:0.73 | Increase in viscosity |
| 2 | PSS$_{176}$-*b*-PNIPAM$_{268}$ | 36.4 | 32.7 | 1.29 | 66.7 | 46.1 | 1.41 | 1:0.83 | Increase in viscosity |
| 3 | PSS$_{176}$-*b*-PNIPAM$_{494}$ | 36.4 | 32.7 | 1.29 | 92.3 | 61.4 | 1.44 | 1:1.53 | Weak gel |
| 4 | PSS$_{96}$-*b*-PNIPAM$_{440}$ | 19.9 | 14.5 | 1.27 | 69.6 | 41.2 | 1.20 | 1:2.3 | Stable gel |

[a]Determined by $^1$H NMR in D$_2$O from monomer conversion.
[b]Determined by gel permeation chromatography in DMF/water buffer 90:10 calibrated against PSSNa standards using a refractive index detector.
[c]Molar ratio of PSS:PEDOT was held constant at 3.5:1.

temperatures above 40 °C. Anecdotally, we have performed more than these 10 thermal cycles in sealed vials without losing the reversible sol-gel transition.

Importantly, the TR-CP can be lyophilized and re-dispersed in aqueous solutions, including aqueous buffers of varying pH, PBS, or cell media, without affecting its reversible sol-gel transition. We re-dispersed the TR-CP in water at pH 7, and on changing the concentration from 3.6 wt% to 5.6 wt%, a negligible increase in $T_{gel}$ was observed to 36 °C (Fig. S10a). However, the gelation time at 37 °C decreased to 23 s, and the storage modulus increased by an order of magnitude to 200 Pa (Fig. S10b), highlighting the possibility to control gelation time and gel stiffness simply by changing the TR-CP concentration in water. TR-CP gels at both concentrations were stable for at least 15 min at 37 °C (Fig. S10c), and for several months in a closed vial at room temperature.

To further test the stability of the TR-CP at 37 °C, the TR-CP gel was formed inside a sealed vial and maintained at 37 °C for 80 days. Under these conditions, the TR-CP remained gelled with negligible changes in volume (Fig. S11). After 80 days, the TR-CP still retained its reversible sol-gel transition, showcasing the robust and stable nature of the gel. The quick gelation time, tunable storage modulus, and good stability under shear and at physiological temperature are all desirable for bioelectronics, particularly to interface with soft tissues.

To trigger a thermo-reversible gelation, we found that covalent functionalization of PEDOT:PSS with PNIPAM was necessary, as a control experiment showed that blending PEDOT:PSS with PNIPAM, as has been done in solid films[23], resulted in only the precipitation of PNIPAM without gelation of PEDOT:PSS. We also found that the block copolymer structure is essential for the sol-gel transition. A random copolymer of PSS with PNIPAM (Fig. S4) used to prepare PEDOT:PSS-*co*-PNIPAM (Fig. S5), with identical mass ratios of all components as in the TR-CP, only displayed a slight increase in viscosity on heating. These control experiments highlight the need for a controlled block copolymer architecture to achieve the desired reversible gelation.

### Electronic properties of the TR-CP

The electronic properties of the TR-CP (Fig. 2) were investigated above and below the LCST using electrochemical impedance spectroscopy (EIS) (Fig. S12a). For the PEDOT:PSS$_{96}$-*b*-PNIPAM$_{440}$ TR-CP, upon increasing temperature, the Bode plot showed a decrease in the impedance magnitude indicative of an increase in conductivity upon gelation (Fig. S12b). To confirm this result, a modified Debye circuit model (Fig. 2c), commonly used for PEDOT:PSS-based hydrogels[24–26], was used to extract the electronic resistance ($R_e$) and ionic resistance arising from the movement of H$^+$ ions in the absence of added electrolyte ($R_i$) (Fig. S12c). We chose the Debye circuit model as it is best suited for studying mixed conduction in materials sandwiched between two electrodes[27,28]. We observed that on increasing temperature, the $R_i$ increased from 86.8 Ω for the liquid to 121.7 Ω for the gel (Table S1). We believe that this increase occurs due to the hydrophobic nature of PNIPAM and the formation of physical crosslinks that

hinder ion diffusion in the matrix. Further, $R_e$ decreased from 41.1 kΩ (liquid) to 22.5 kΩ (gel) on increasing temperature (Table S1), which is presumably responsible for the significant impedance drop observed in the low-frequency regime (Fig. S12b). A two-point probe measurement confirmed a similar decreasing trend in resistance, from 38.1 kΩ to 20.1 kΩ upon gelation, indicating that the Debye model accurately reflects the physical and chemical properties of the TR-CP gel. The observed reduction in electronic resistance after gelation suggests the formation of a more electronically conductive pathway in the gel, likely resulting from closer packing of PEDOT chains. Despite this reduction, the $R_e$ of the formed gel was nearly two orders of magnitude higher than previously reported conductive hydrogels characterized with the same setup and model (~200 Ω)[17,25]. In order to attain comparable $R_e$, we investigated the effect of the PEDOT loading and TR-CP concentration on $R_e$, as we expect these parameters to increase the potential for PEDOT chains overlap and therefore achieve higher electronic conductivity.

The SS:EDOT molar ratio was changed to 3.5:1, 2.2:1, or 1.75:1 by changing the loading of EDOT during the oxidative polymerization. SS:EDOT = 3.5:1 was chosen as the lower limit, as decreasing the loading of EDOT further only resulted in a slight increase in viscosity above the LCST, not gelation. At ratios higher than 1.75:1, the dispersions were not homogeneous at room temperature. We found that the 3.5:1 ratio showed the lowest impedance (Fig. 2a), which we presume is due to the most homogenous dispersion of PEDOT chains. From the equivalent circuit model (Fig. 2c), the ionic ($\sigma_i$) and electronic conductivity ($\sigma_e$) were calculated to be 34 mS cm$^{-1}$ and 0.05 mS cm$^{-1}$, respectively (Fig. 2b). The TR-CP gel was incorporated into an electric circuit by depositing a liquid drop on a glass slide heated to 37 °C, gelling the TR-CP, demonstrating its ability to close a simple LED circuit upon gelation (Fig. 2d).

Lastly, we varied the concentration of the TR-CP, thereby optimized with a SS:NIPAM:EDOT ratio of 3.5:16:1, by reconstituting the lyophilized TR-CP in a phosphate-buffered saline (PBS) solution at different concentrations (Fig. 2e). As the concentration was increased from 3.6 wt% to 5.6 wt%, the 2-point probe DC conductivity increased from 2.1 mS cm$^{-1}$ to 7.1 mS cm$^{-1}$. At 7.6 wt%, at which concentration the solution was very viscous and heterogeneous, the conductivity decreased to 5.5 mS cm$^{-1}$. The Nyquist plots for 3.6 wt% (Fig. S13a) and 5.6 wt% (Fig. 2f) displayed a smooth semi-circle, indicating good dispersion and homogeneity of the gel, while that of the 7.6 wt% TR-CP had irregular signals (Fig. S13b), further confirming that the gel was heterogeneous, leading to poor contact with the electrode. The fitting values of all circuit elements are tabulated in Table S2, and calculated electronic and ionic conductivities are reported in Table S3. The highest conductivity recorded was for 5.6 wt% TR-CP (PSS:PNIPAM:PEDOT ratio of 3.5:16:1) in PBS: $\sigma_i = 200$ mS cm$^{-1}$ and $\sigma_e = 14$ mS cm$^{-1}$ from EIS (Fig. 2f) and $\sigma = 10.4$ mS cm$^{-1}$ from 2-point probe, which is on par with other previously reported conductive hydrogels based on PEDOT:PSS and formulated for bioelectronic interfaces[16,23,25]. We note that the ionic conductivity of the TR-CP gels is

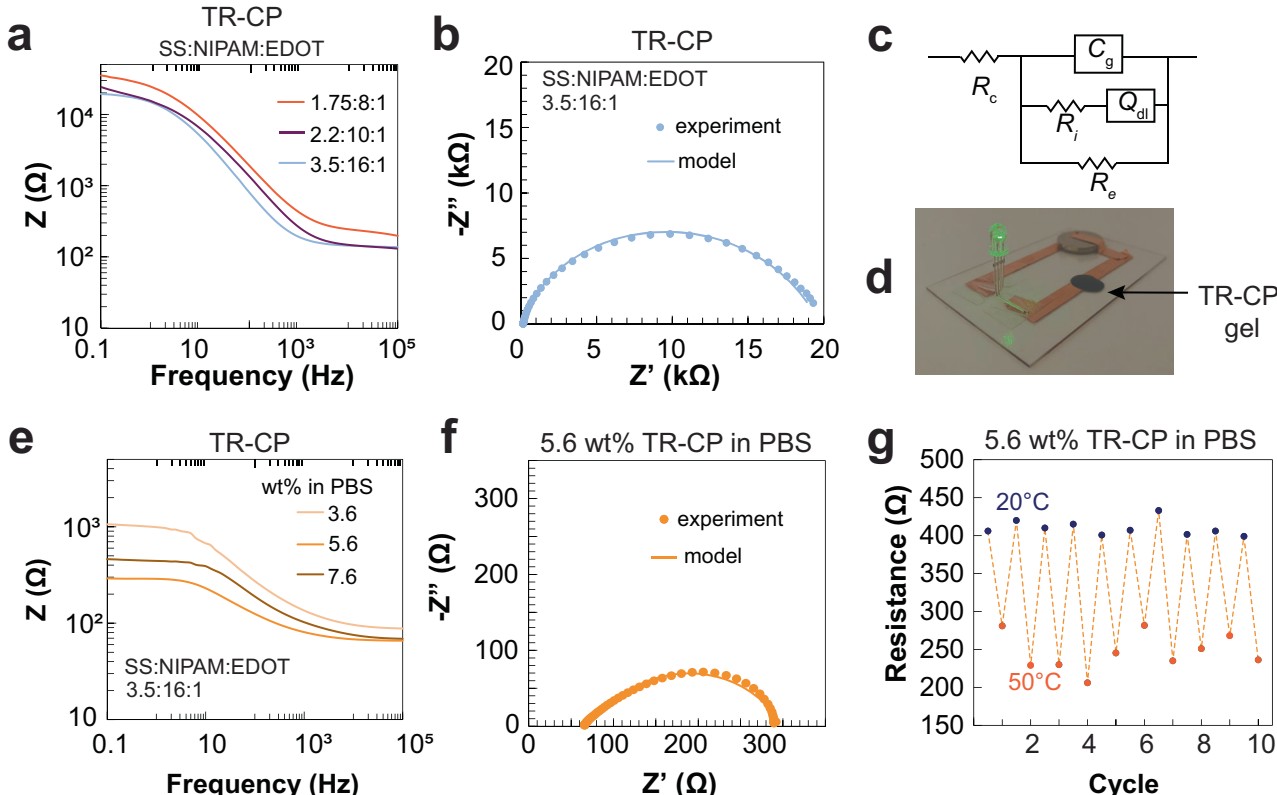

**Fig. 2 | Electronic properties of the PEDOT:PSS$_{96}$-*b*-PNIPAM$_{440}$ TR-CP. a** Bode plot obtained by EIS showing the effect of PEDOT loading on the impedance of the gels (3.6 wt% in DI water). The TR-CP with SS:NIPAM:EDOT ratio of 3.5:16:1 showed the lowest impedance. **b** Nyquist plot of the TR-CP with SS:NIPAM:EDOT ratio of 3.5:16:1. **c** Equivalent circuit model used to fit the EIS data and obtain ionic ($R_i$) and electronic ($R_e$) resistance. $R_c$ is the resistance from the set-up, $C_g$ is a constant phase element and Q$_{dl}$ is a double-layer capacitor. **d** Photograph showing an LED circuit completed with a TR-CP gel (SS:NIPAM:EDOT ratio of 3.5:16:1, 3.6 wt% in DI water). **e** Effect of TR-CP concentration in PBS (pH = 4) on the impedance magnitude. The TR-CP was lyophilized and re-dispersed at given concentrations in PBS. **f** Nyquist plot for the TR-CP at 5.6 wt% in PBS (pH = 4). This formulation shows the highest conductivity among all the TR-CP samples. **g** Change in resistance (2-point probe DC measurements) as a function of temperature of the TR-CP (5.6 wt% in PBS, pH = 4) for a sample volume of 0.4 cm$^3$.

much higher in PBS than in DI water, consistent with a higher concentration of mobile ions. Finally, the reversibility of the electronic properties of this TR-CP formulation was evaluated by repetitively (10×) cycling the temperature between 50 °C (gel) and 20 °C (solution) while monitoring the electrical resistance. The gel displayed stable resistance values in both states upon multiple temperature cycles (Fig. 2g).

## Microstructure and gelation mechanism

To characterize the microstructure and mechanism of gelation, microscopy and X-ray scattering techniques were employed. To elucidate the nanoscale structure of the TR-CP and what prompts it to form a gel, we performed cryogenic transmission electron microscopy (cryo-EM) (SI Section 2.4). The samples were vitrified at two different temperatures, 25 °C and 45 °C, and two different concentrations, 1.8 wt% (no gelation observed, Fig. S14) and 3.6 wt% (gelation observed; Fig. 3a–d). For samples vitrified below the LCST (Figs. 3a, c and S15a), the TR-CP at both concentrations exhibited structures with two different length scales: small spherical/ovoid particles with a radius of 4.7 ± 0.6 nm, sometimes assembled in a flexible chain—reminiscent of filomicelles—over 20 nm, and rod-like lamellar fringes predominantly outside these nanoparticles with close to 1.6 nm spacing (Fig. S16, S17a, S17b). Due to their high contrast, we expect that the spherical particles are comprised of PEDOT-rich regions, surrounded by PSS-*b*-PNIPAM. The lamellar fringes are likely associated with a semi-crystalline PEDOT phase. The small size of these particles in solution likely explains the excellent colloidal stability of the dispersion and their ability to be lyophilized and redispersed. For samples vitrified at 45 °C (above the

LCST), cryo-EM showed significantly different structures depending on concentration. In the 1.8 wt% sample, we observed the formation of isolated spherical structures spaced about 50 nm apart (Fig. S15b and S17c), while the 3.6 wt% samples showed the aggregation of the spherical particles leading to a bi-continuous phase with a mesh size of about 50 nm (Figs. 3b, d and S17d). In addition, cryo-EM micrographs of 1.8 wt% samples vitrified at 45 °C also showed the presence of lamellar fringes with ~1.6 nm spacing and small particles (~5 nm) in these samples (Figs. S18 and S19). We confirmed that the lamellae visible in cryo-EM images are not due to artefacts from the oscillating contrast transfer function of the microscope by demonstrating that the defocus value does not affect the lamellae spacing (Fig. S18). These results show that although increasing the temperature caused the aggregation of the TR-CP colloids in both samples, only the 3.6 wt% samples met the criteria needed for gel formation. This observation is consistent with the visual inspection of the TR-CP and rheology, identifying 3.6 wt% as the lower limit of concentration for gel formation. For higher concentrations (such as 5.6 wt% TR-CP), there is a possibility of the formation of a "pre-gel" state (Fig. S20) before the formation of a fully interconnected network.

To confirm our observations from microscopy, small-angle X-ray scattering (SAXS) at 20 °C and 40 °C in solution was used to study the colloidal particle self-assembly, gel structure, and gelation mechanism (Fig. 3e, additional details in SI Section 2.5). A core-shell sphere model fitting[29] revealed that the TR-CP colloids in solution below the LCST of 3.2 nm PEDOT cores surrounded by a 0.9 nm PSS-PNIPAM shell, very close to the size of the spherical/ovoid nanoparticles seen by cryo-EM (Fig. S21 and Table S4). A strong structure factor peak S(q)

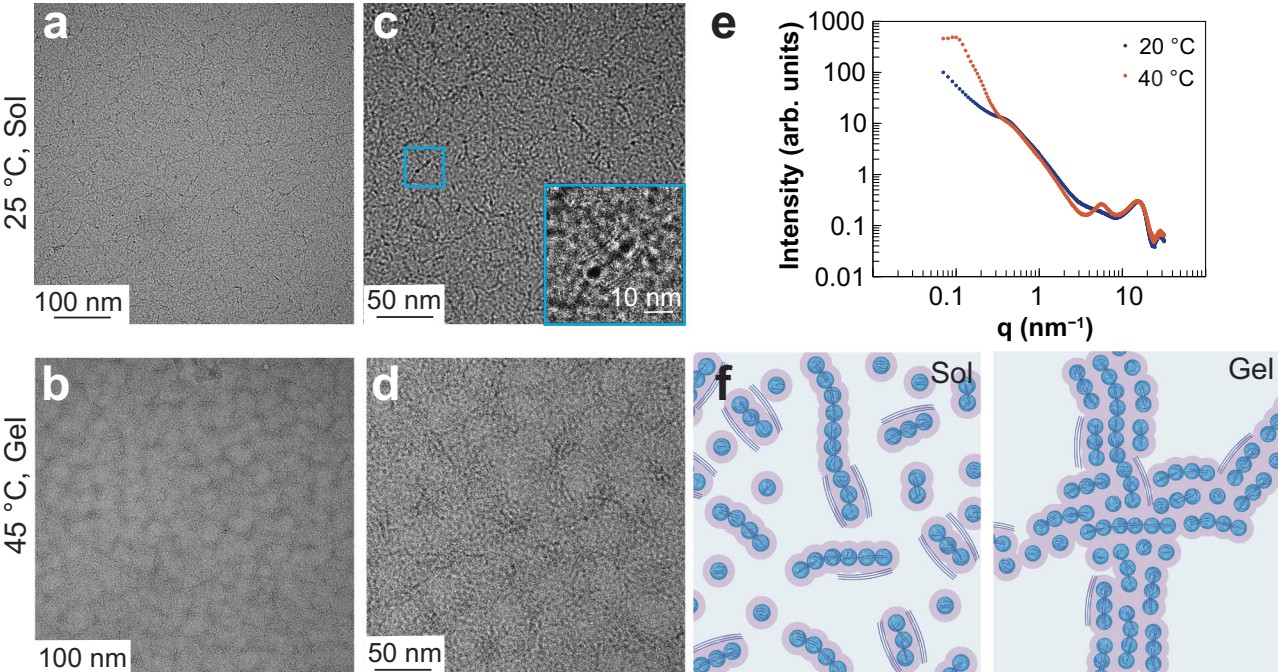

**Fig. 3 | Characterizing the microstructure and gelation mechanism of the PEDOT:PSS$_{96}$-b-PNIPAM$_{440}$ TR-CP (PSS:PEDOT molar ratio = 3.5:1).** TEM micrographs were collected from two different sets of samples. To ensure reproducibility, multiple independent spots were imaged on each TEM grid **a**, **c**. Cryo-EM images of 3.6 wt% TR-CP liquid below LCST, inset: lamellar fringes outside the colloidal particles. **b**, **d** Cryo-EM images of 3.6 wt% TR-CP above LCST. **e** Temperature-dependent small angle X-ray scattering (SAXS) of 3.6 wt% TR-CP in solution in water. **f** Schematic of the self-assembled TR-CP colloids and proposed mechanism of gelation. Created in BioRender. Damani, V. (2025) https://BioRender.com/v16a157, https://BioRender.com/xe9umwp.

appeared in low q range (centered at 0.102 nm$^{-1}$). These features indicate that after gelation, the particles aggregated to form large, ordered structures with a center-to-center distance of ~60 nm, in line with the cryo-EM results at 45 °C. The peak centered at 5.6 nm$^{-1}$ is attributed to the intermolecular interactions from the PNIPAM. Two peaks in the high q region ($q = 15$ nm$^{-1}$ and $q = 28$ nm$^{-1}$) are attributed to the sample holder cells and water background, and are excluded from the fitting (Fig. S21d, e). As the temperature increases above gelation, both the core radius and the shell thickness of these spherical nanoparticles slightly increase to 3.9 nm and 1.3 nm, respectively. The interparticle distance (effective radius) of the nanospheres decreased from 12.2 nm to 5.8 nm above the LCST, consistent with the transition into a gel state through particle aggregation. We note that the SAXS did not capture the presence of the PEDOT lamella seen by cryo-EM due to their low concentration, limited number of stacked chains (i.e., small crystal size), and overlap with background scattering.

Based on these experiments, the following mechanisms for the formation of the particles and the gelation are proposed (Fig. 3f). During the oxidative polymerization of EDOT in the presence of PSS-b-PNIPAM (below the LCST), PEDOT$^+$ chains grow and complex with the PSS$^-$ block. This polyelectrolyte complex is likely more hydrophobic than PNIPAM, which results in the formation of spherical/ovoid particles with a PEDOT:PSS core and PNIPAM shell. The presence of PNIPAM in the shell is consistent with results from the surface elemental analysis by X-ray photoelectron spectroscopy (XPS), showing that the ratio of N:S is always higher (6.34 ± 0.36) than the theoretical N:S ratio of 4.5:1 (Fig. S22). In some cases, these particles are assembled in worm-like chains, likely through the polymerization of longer PEDOT chains within the core of the colloidal particles that forced the fusion of the spherical/ovoid particles. Occurrences of lamellar fringes outside the worm-like micellar chains were seen in cryo-EM micrographs, implying that some PEDOT also polymerized outside the shell and was electrostatically bound to the colloidal particles. Post-synthesis and above the LCST at a concentration of 3.6 wt%, the colloidal particles aggregate into a bi-continuous network while generally maintaining (or perhaps slightly increasing) their overall size, which leads to gelation. This aggregation can be attributed to inter-molecular hydrogen bonding of the PNIPAM domains above the LCST while the PEDOT:PSS core remained unchanged, consistent with results from Raman spectroscopy (SI Section 2.7, Fig. S23 and Table S5). Overall, the block copolymer architecture of the PSS-b-PNIPAM was essential to achieving reversible gelation. This architecture ensured the self-assembly to colloids with the PNIPAM largely located at the surface, which ensured that the thermo-responsive PNIPAM was available for intermolecular hydrogen bonding above its LCST, leading to the formation of a colloidal network via non-covalent crosslinking.

## In vitro and in vivo cytocompatibility of TR-CP and TR-CP/alginate composites

Biocompatibility is an important property of materials intended for bioelectronic and tissue engineering applications. The cytocompatibility of the TR-CP was assessed using the modified ISO 10993-5 protocol (SI Section 2.8, 2.9)[15]. In addition to the pristine TR-CP, we also studied the cytocompatibility of TR-CP/alginate composites, as a permanently crosslinked network would allow for more control in experiments (no need to constantly maintain T > LCST) and would allow for reliable gel identification in the in vivo studies. For these studies, we adjusted the TR-CP to a neutral pH. Characterization of the rheological (Fig. S10a–c) and electronic properties (Fig. S24 and Tables S6–S7) shows that the TR-CP behaves similarly under this pH and retains its sol-gel transition when reconstituted in cell media (Fig. S25). Additionally, the gels can withstand a range of pH and retain their sol-gel transition below 37 °C even under basic conditions, up to pH = 12 (Fig. S26), highlighting that this material can be used under a range of different physiological conditions.

To assess the in vitro cytotoxicity by indirect contact, alamarBlue™ assays were performed on extracts from the 3.6 wt% TR-CP gels (pH = 7, reconstituted in sterile cell media). L929 fibroblasts

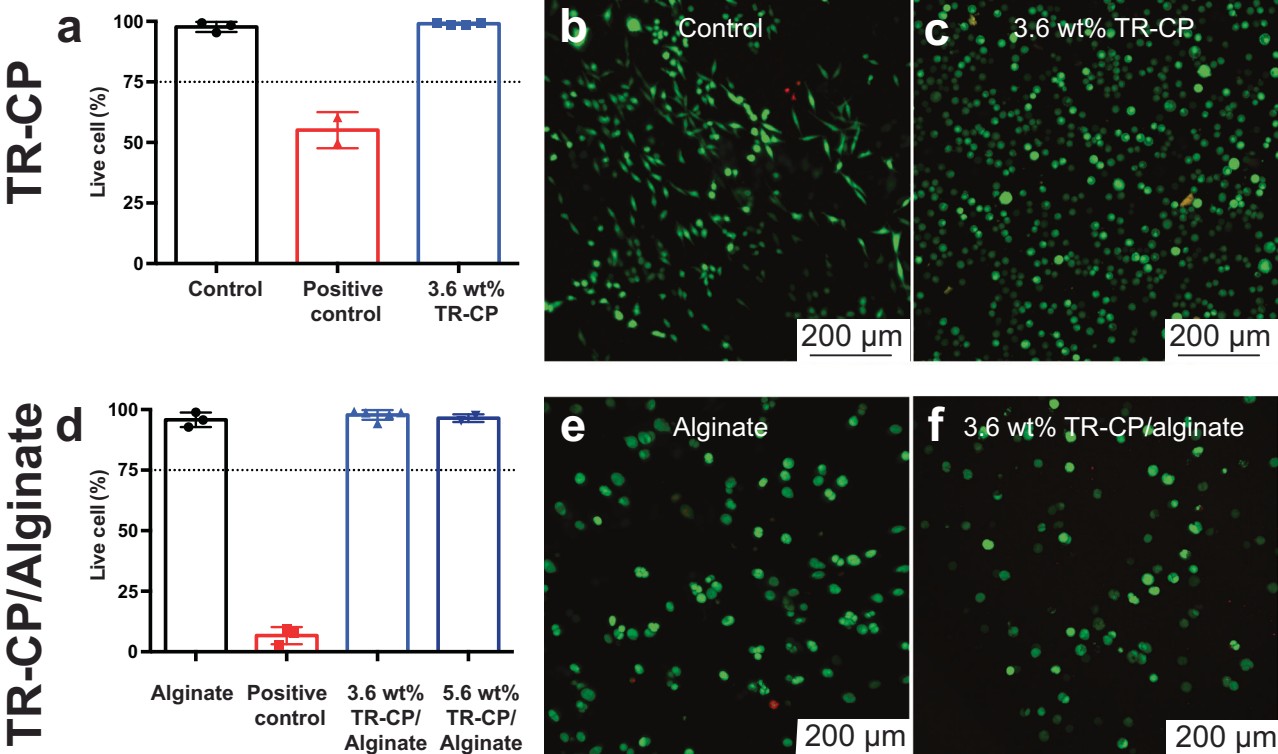

**Fig. 4 | Direct cytocompatibility assays with L929 fibroblast cells using fluorescence imaging with live/dead stains.** Live cells fluoresce green (calcein-AM) while nuclei from dead cells fluoresce red (ethidium homodimer-1). **a** Live cell % for 3.6 wt% TR-CP in PBS (pH = 7) compared to controls. The cells were incubated for 96 h for each group. Data is presented as mean ± std. deviation of biological replicates. $n = 3$ for control, 2 for positive control, 4 for 3.6 wt% TR-CP. **b** Control: cells after incubating with control collagen gel; **c** 3.6% TR-CP: cells after incubating with TR-CP. **d** Live cell % for TR-CP/alginate composite. The cells were incubated for 48 h for each group. Data is presented as mean ± std. deviation. $n = 3$ for alginate, 3 for positive control, 5 for 3.6 wt% TR-CP, 2 for 5.6 wt% TR-CP. **e** Control: cells after incubating with control alginate gel; **f** 3.6% TR-CP/alginate: cells after incubating with 3.6% TR-CP in alginate gel.

were cultured in the extracts and incubated for either 2 days or 7 days. Percent viability was calculated with respect to a sterilized cell culture media control. The day 1–5 extracts from the TR-CP all supported L929 fibroblasts well with high viability for both the 2-day (>84%, Fig. S27a) and 7-day cultures (>89%, Fig. S27b).

Next, a direct contact in vitro assay was performed (Fig. 4a–c) by suspending L929 fibroblasts in a media-reconstituted TR-CP dispersion before gelling. Collagen was used as a control, and ethanol-treated collagen was used as a positive control (Fig. S28a). We saw high cell viability (>95%, Fig. 4a) in the TR-CP gel. The control displayed larger cell spreading (Fig. 4b), potentially due to the stable gelation and structural integrity of the collagen, whereas cell culture in the pure 3.6% TR-CP resulted in less cell adhesion and spreading due to fluctuations in temperature during cell staining procedures (Fig. 4c). In preparation for in vivo studies, the TR-CPs were then incorporated into alginate precursor and subsequently chemically crosslinked to form a permanently crosslinked TR-CP/alginate composite (Figs. 4d–f and S29). Direct contact viability was tested for TR-CP/alginate with two loadings of TR-CP in the composite (3.6 wt% and 5.6 wt%). Alginate treated with ethanol was used as positive control (Fig. S28b). At both concentrations, no significant difference in cell viability was observed (Fig. 4d) between the control alginate gel (Fig. 4e) and TR-CP/alginate gels (>95%, Figs. 4f and S29). The positive cell outcomes of the TR-CP are consistent with prior reports of PEDOT:PSS composites containing PNIPAM[23,30,31], and confirm the high in vitro cytocompatibility of the TR-CP. We observed that in the collagen control group, the cells spread and adopted spindle-like shapes (Fig. 4b), while in the 3.6 wt% TR-CP and the TR-CP/alginate composite the cells exhibited less spreading and had a rounded shape (Fig. 4c). These differences are consistent with previous

reports[32,33], and explained by the fact that the collagen is stiffer than the TR-CP and TR-CP/alginate. The rheological and electronic properties of TR-CP/alginate are summarized in Fig. S30–S31. The samples and their controls for cytocompatibility are listed in Table S8.

To test the in vivo cytocompatibility of the TR-CP, permanently crosslinked 5.6 wt% TR-CP/alginate composite gels and control alginate gels were implanted subcutaneously into Sprague Dawley rats (Fig. 5a–c). The gels were explanted at 1 and 2 weeks post implantation to check for acute inflammatory response. At both time points, the rats showed no signs of distress, and the wound sites did not exhibit any signs of infection or hematoma. In both treatment groups, H&E staining revealed the intact gels surrounded by a layer of fascia residing beneath the muscle layer of the skin (Fig. 5d, e, g, h). The connective tissue surrounding the TR-CP/alginate gels intimately interfaced with the gel (Fig. 5f), though the thicker layer of tissue surrounding the TR-CP gel indicates a slightly increased inflammatory response compared to control. As others have observed, an increased inflammatory response is possible when conducting polymers are implanted[34,35]. There were also signs of cell infiltration to the site of the gel, indicated by the presence of microvasculature in the surrounding tissue (Fig. 5i). We also observed differences in maintenance of the gel volume between the two materials. Alginate is easily degraded in the body due to an ion exchange mechanism that displaces the cross-linking divalent cations[36]. This degradation is characterized by a reduction in the volume of the gel after implantation. The addition of TR-CP to the gel prevented this breakdown, as the volume of the gel was maintained (Fig. 5e). Similarly to PEDOT:PSS[19], we believe that the TR-CP is non-biodegradable and likely disrupted ion flow, thereby preventing the decrosslinking of the alginate. Despite the increased cellular activity, the TR-CP did not cause any necrosis in the local skin

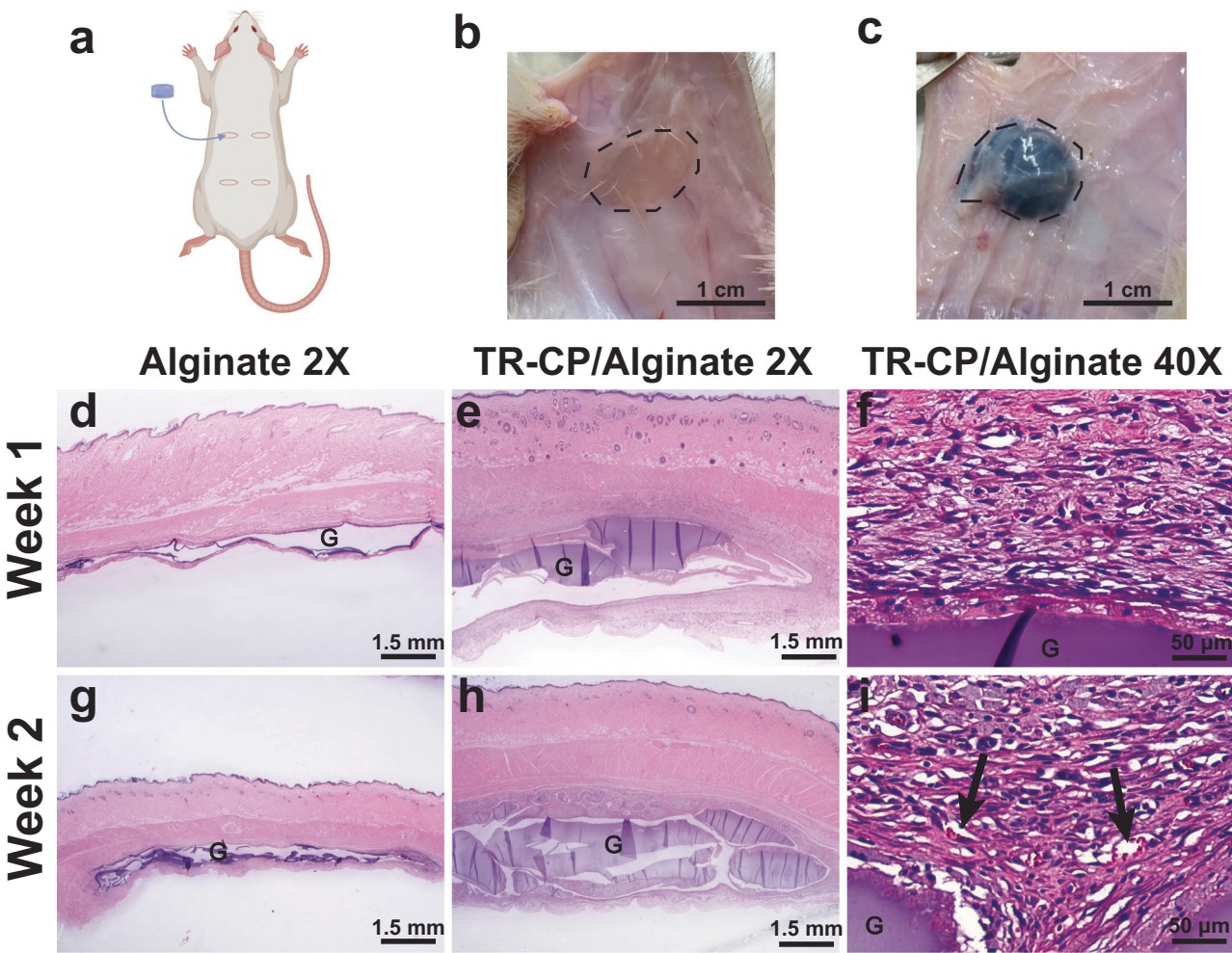

**Fig. 5 | Subcutaneous implantation in Sprague Dawley rats. a** Diagram of the placement of the subcutaneous pockets and gel implants. Animal sample size was $n = 2$, each animal had $n = 4$ samples implanted for $n = 4$ per condition. Created in BioRender. Daso, R. (2025) https://BioRender.com/6gg4lm0 Photos of **b**. The alginate gel control and **c**. TR-CP/alginate composite gel in their subcutaneous pocket 2 weeks post implantation. **d–i** Hematoxylin-eosin staining of skin sections with implanted gels at 1 and 2 weeks post implantation where G indicates gel area visible as dark purple color and arrow points to microvasculature observed in the TR-CP/alginate samples, which appear as small ovoid vessels containing red-stained erythrocytes. Scale bar **d, e, g, h** = 1.5 mm, **f, i** = 50 μm. The micrographs pictured (**d**–**i**) were 1 of 2 biological samples per gel per time point.

environment. This study suggests the TR-CP/alginate composite gels have minimal negative effects to the overall tissue function, and while unoptimized, have potential as conductive biological interfaces. For the scope of this study, we have not investigated the effects of repeated thermal cycling on cells and tissues, but mechanical studies have shown that the properties of the materials are stable upon cycling.

**Processing of the TR-CP and demonstration as a reusable s-EMG electrode**

To explore the versatility of the TR-CP, we studied its processability by deposition on warm substrates or injection in warm fluids. First, we drop-casted 5.6 wt% TR-CP onto a glass slide heated to 37 °C (Fig. 6a, Supplementary Movie 2). The gel formed instantly on contact with the glass and assumed a hemispherical shape. The TR-CP gel was sliced with a razor blade, and the two halves maintained their shape well when kept at 37 °C. The gel was then allowed to cool to room temperature and was re-formed on heating the glass slide to 37 °C, thus displaying a self-healing behavior. Further, we investigated the injectability of TR-CPs into a warm, viscous alginate precursor solution (noncrosslinked) maintained at 37 °C (Fig. 6b and Supplementary Movie 3). The injected TR-CP gelled instantly, displayed minimal spreading, and stayed gelled as long as the temperature was maintained above 35 °C

(Fig. S32). This behavior may be harnessed in future studies to make injectable conduits for electrical stimulation, 3D prints of functional materials and cell structures via support printing[37,38], removable and conductive scaffolds for tissue engineering, or injectable microelectrode arrays for sensing and recording[8,11,39,40].

Lastly, to demonstrate an application for the TR-CP, we developed a conformal and reusable surface electromyography (s-EMG) epidermal electrode (Fig. 6c). The TR-CP (5.6 wt% in PBS) was drop-casted into a soft PDMS reservoir and heated until gelation to form electrodes. The TR-CP device was adhered to skin using a transparent dressing, along with a heating patch to ensure a stable temperature throughout the experiment. We note that the heating patch may not be necessary in practical applications as the TR-CP gels on contact with warm surfaces, including skin. While conformal, the TR-CP is only weakly adhesive to skin, and the dressing was therefore useful to maintain the device in place. We are currently working on a new formulation that would be self-adhesive and prevent the use of a potentially irritable tape. The device was used to record fist closing and opening movements (Fig. 6d and Supplementary Movie 4). We observed that in comparison to commercial electrodes, the TR-CP device displayed a higher signal amplitude (250×) with minimal parasitic noise (Fig. 6e). We attributed this performance enhancement to

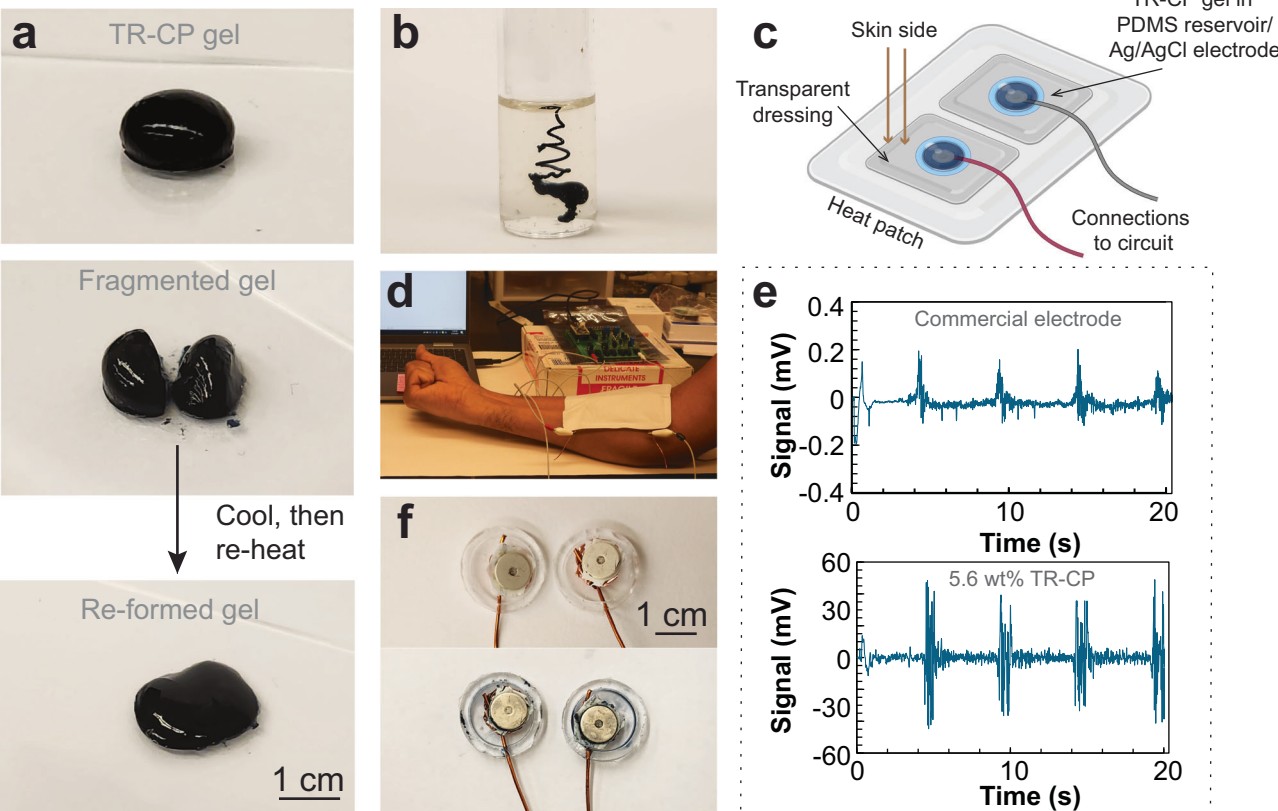

**Fig. 6 | Processing and example of application of the TR-CP. 5.6 wt% TR-CP in PBS at pH = 4 was used for all the experiments. a** Self-healing of the TR-CP. The TR-CP gel is formed by deposition on a heated glass slide, then cut and allowed to re-form by cooling slightly and re-heating. The gels took ~25 s to heal for a volume of 0.6 mL. **b** Picture of the TR-CP injected using a 21-gauge needle into a pre-heated alginate solution (1.8 wt% alginate in DI water, $T$ = 37–40 °C). **c** Schematic of the device for s-EMG measurements. Created in BioRender. Damani, V. (2025). https://BioRender.com/qp0ttib **d** Picture of s-EMG set-up with TR-CPs, showing the connections and fist closing movement. **e** Change in signal amplitude as a function of time, where each peak corresponds to fist closing and opening. **f** Picture of s-EMG electrodes showing the Ag/AgCl connections and PDMS reservoirs for the TR-CP before and after use.

the mixed ionic-electronic conductivity of the TR-CP and shape-conforming behavior due to its soft and adaptable mechanics. The low modulus of this soft electronic interface prevented any discomfort on skin and provided good conformal contact throughout the measurement. Post-recording, the electrodes were cooled down and TR-CP liquid was extracted, thus demonstrating the reusability of the TR-CPs (Fig. 6f) and their potential for use as shape-conforming bioelectronic interfaces which could be harnessed for measurements on non-uniform surfaces such as hairy skin.

## Discussion
In this study, we developed intrinsically thermo-responsive and conductive polymers (TR-CPs) with a fully reversible sol-gel transition to circumvent the need for covalent and irreversible crosslinking in conductive hydrogels. We leveraged the self-assembly of block polyelectrolyte complexes to achieve stable and well-defined colloids. Above the transition temperature (35–36 °C), dispersions of the TR-CP undergo a very rapid (<1 min) and isovolumetric gelation under a range of pH and ionic environments, resulting from the aggregation of the colloids into a bi-continuous network. These gels are electrically conductive and cytocompatible, and therefore amenable to applications in bioelectronics, which we demonstrated in a conformal and reusable surface electromyography device.

The molecular design introduced in this study may inform the development of organic conductors for applications in bioelectronics and beyond. Block copolymers could be used to tune the viscosity of PEDOT:PSS derivatives for injection 2D and 3D printing or adding other functionalities than thermo-responsiveness, such as

adhesiveness, analyte selectivity, and biodegradation, which we will study in the future. From a bioelectronics perspective, we envision that the TR-CP will find use in a range of different applications. For example, their high conformability and hydration would improve interfacing with uneven and oddly shaped substrates, such as hairy skin, scalp, and irregularly shaped wounds. For in vitro bioelectronics requiring phenotyping studies after encapsulation, the de-crosslinking of the gels upon cooling may enable easy recovery of the biological substrate while preventing tissue or cell damage. Lastly, the thermo-response also enables injectability from very fine needles without applying excessive shear stress, which may be beneficial for injectable scaffolds, electrodes, and bioprinting with high precision.

## Methods
### Materials
Diethyl ether, carbon disulfide, sodium thiomethoxide, ethyl acetate, α-Bromophenylacetic acid, hydrochloric acid, sodium chloride, n-hexane, 4,4′-Azobis(4-cyanopentanoic acid) (ACVA), 3,4-ethylene-dioxythiphene (EDOT), sodium persulfate, iron (III) chloride, sodium alginate (PRONOVA ® UP MVG), barium chloride, sodium hydroxide were obtained from Sigma Aldrich and used without further purification. Sodium styrene sulfonate (NaSS) was purchased from Sigma Aldrich and recrystallized from ethanol/water. *N*-isopropylacrylamide (NIPAM) was purchased from Sigma Aldrich and recrystallized from *n*-hexane. Dowex Marathon C (acidic resin) and Lewatit Ion Exchange (basic resin) were used for purification wherever specified. Distilled water filtered using a Milli-Q purification system was used throughout.

### Nuclear magnetic resonance (NMR)

[1]H NMR spectra were obtained on a Bruker 400 MHz spectrometer. All the crude reaction mixtures and polymers were dissolved in $D_2O$ at a concentration of ~10 mg/mL. All measurements were taken at room temperature. Data was analyzed using MestreNova x64.

### Gel permeation chromatography (GPC)

GPC was performed using a Tosoh HLC-8420 GPC EcoSEC LC system running in 0.1% wt LiBr in 90% DMF and 10% DI water (0.8 mL min[−1]), using two PSS GRAM Analytical Linear (10 μm, 8 * 300 mm) columns and a refractive index detector. Number-average ($M_n$) and weight-average ($M_w$) molecular weights and dispersity (Đ) were determined by calibration against narrow dispersity polystyrene sulfonate standards (purchased from Polymer Standards Service).

### Rheology

Rheological characterization was performed on the AR-G2 rheometer from TA Instruments, using 40 mm parallel plate geometry for TR-CPs and 25 mm parallel plate geometry for alginate and TR-CP/alginate gels. Data was collected using the TRIOS software.

**TR-CP**. Variable-temperature rheology was used to obtain the mechanical properties of TR-CPs. Based on the viscosity and volume (0.6–0.8 mL) of samples, parallel plate geometry with radius of 40 mm was chosen. All measurements were performed in the temperature range of 20 °C to 50 °C, with solvent trap covers to minimize dehydration effects. The storage modulus or elastic modulus (G′) and loss modulus or viscous modulus (G″) and tan δ (G″/G′) were characterized using a temperature ramp from 20 °C to 50 °C, and crossover point (tan δ = 1, G″ = G′) was recorded. For this study, the temperature was increased gradually at a constant rate (5 °C/min), with angular frequency (ω) = 10 rad/s and at 5% strain. Further, using the same experimental parameters, a time sweep was performed at 37 °C to determine the thermo-response at physiological temperature. To observe reversibility and cycling stability, G′ and G″ were studied over 10 heat-cool cycles from 20 °C to 50 °C, using the same parameters listed above. Finally, a frequency sweep was performed from 0.01 rad/s to 100 rad/s at 37 °C, and the G′ and G″ were recorded to observe the frequency-dependent behavior.

**Alginate and TR-CP/alginate composites**. All measurements were taken on the AR-G2 rheometer from TA instruments, using a parallel plate geometry with diameter of 25 mm for alginate gels and 20 mm for TR-CP/alginate gels. Sample thickness was controlled between 800 and 900 μm. First, amplitude sweeps were used to determine the stable or linear region for the gels. The amplitude sweeps were performed at 25 °C by varying the oscillation strain % from 0.0001% to 100%. To record the thermo-response, storage modulus or elastic modulus (G′) and loss modulus or viscous modulus (G″) were characterized using temperature ramps from 20 °C to 50 °C for alginate and 25 °C to 50 °C for TR-CP/alginate. First, a heating ramp was performed, followed by a cooling ramp to observe the reversibility of thermo-response. The temperature was changed gradually at a constant rate (2.5 °C/min), with angular frequency (ω) = 10 rad/s and at 0.005% strain for alginate and 0.05% strain for TR-CP/alginate.

### Electrochemical impedance spectroscopy (EIS)

EIS was performed on Metrohm Autolab PGSTAT128N potentiostat/galvanostat. The impedance was scanned from 0.1 to 100,000 Hz at a DC offset of 0 V and an AC amplitude of 10 mV. The equivalent circuit model fitting was completed on the NOVA 2.5 software.

All electronic characterization was done using the custom two-electrode set-up shown in Fig. S12a. Here, we have used a two-point probe for our measurements. The four point probe method is not reliable for such soft hydrogels, because the thickness of the sample

changes as the probe presses down, which would change the conductivity calculation and only reflects surface conductivity, which is not as relevant. On the other hand, the method we have previously developed using two electrodes in a cuvette allows us to maintain precisely the geometry for most accurate bulk measurements[17,25,41]. For the electronic measurements on samples with variable PEDOT concentration, $1 cm^3$ of TR-CP was used. For all other electronic measurements, sample volumes and solvent conditions are as specified in captions.

The Debye model[28] was used to study the ionic and electronic conduction in the bulk of materials, using the two-electrode system, which is employed through sandwiching the hydrogel between Cu electrodes. In our work, the goal of applying EIS is to track the evolution of ionic and electronic conduction along with the change in the hydrogel composition and physical state. Hence, the two-electrode set-up was chosen, and the Debye model was applied. In the modified Debye circuit model that we have employed, $R_c$ is the resistance from the set-up, and thus, remains similar for the liquid and gel. $C_g$ is the geometric capacitance of the conductive hydrogel. $Q_{dl}$ is the constant phase element representing a non-ideal double-layer capacitance. These phase elements are incorporated to account for any heterogeneity in the sample or irregularities in capacitance.

### Two-point probe measurements

To measure the resistance of the TR-CP samples before and after the gelation, an INNOVA multimeter was used, and the ohmmeter model was selected. The set-up used is shown in Fig. S12a.

### Cryogenic transmission electron microscopy (Cryo-EM)

The self-assembly and gelation of the TR-CPs were captured using cryo-EM. Quantifoil Holey Carbon Grids, 300 mesh, with 1 μm hole size and 2 μm spacings (Quantifoil MicroTools, Jena, Germany) were used for vitrification of TR-CP dispersions with 1.8 wt% (mechanical properties shown in Fig. S14) and 3.6 wt% concentrations in water. The grids were glow-discharged in a plasma cleaner for three cycles, each cycle lasting about five minutes with the carbon side of the grids facing the plasma. FEI Vitrobot (FEI Company, Hillsboro, OR) was used for the vitrification of the TR-CP-coated grids. The 25 °C samples were vitrified with the Vitrobot chamber at room temperature and relative humidity of 100%. We minimized the potential for artefacts from water evaporation by preparing samples under 100% relative humidity, to maintain sample hydration during blotting and prior to plunge freezing. Although ice formation can sometimes occur, we prepared 3–5 grids per temperature condition, and grids showing evidence of crystalline ice were excluded from imaging. The vitrification was done using filter papers mounted to the blotting pads with 6 s of total blotting time. Cryogen ethane was used for the vitrification of the samples, and the grids were quickly moved to liquid nitrogen containers after the initial vitrification. For 45 °C samples, the Vitrobot chamber was heated to 45 °C while the solution was kept at 37 °C. We chose 37 °C for solution temperature as it allows the formation of vitreous ice layers, which are thin enough for TEM imaging while maintaining the gelation behavior of the block copolymer. The gelation of the TR-CP at higher temperatures led to thick layers that were challenging to image. For 45 °C samples, the Vitrobot chamber was heated to 45 °C while the solution was kept at 37 °C. We used solutions at 37 °C, as they allow the formation of vitreous ice layers that are thin enough for TEM imaging (<100 nm) while maintaining the gelation behavior of the TR-CP. The Quantifoil grids were also kept at 45 °C for at least 10 min before the blotting to reduce the chance of temperature loss during pipetting and blotting for gel samples. The vitrification was done using filter papers mounted to the blotting pads with 12 s of total blotting time. High-resolution cryogenic experiments were performed on the FEI Krios at the Huck Institute of the Life Sciences at Pennsylvania State University. The measurements were conducted using a 300 kV electron source and a Falcone 4 direct electron detector in

linear mode. The nanoprobe mode of the microscope with a 70 μm C2 aperture, with a spot size of 5, was employed to collect the TEM images. All cryo-EM micrographs were collected with an electron dose of <50 e/Å$^2$ and an electron rate of <13 e/Å$^2$s. Cryo-EM micrographs were collected at two different magnifications of 14,000x and 47,000x with a pixel size of 1 nm and 0.14 nm, respectively. Binning of 1 (no camera pixel binning) was used for the images with 4096 × 4096 pixels. DigitalMicrograph software was used for the analysis of cryo-EM micrographs, including line scan and Fast Fourier Transform (FFT) analysis. Fiji software and radial integration plugin were used for radial integration of 2D FFT plots[42]. In FFT micrographs, the spatial frequency/reciprocal space vector ($g$) is inversely proportional to the real space distance ($d = 1/g$), as conventionally defined in microscopy analyses. On the other hand, the scattering vector in X-ray scattering results is related to real space distance as $d = 2\pi/q$. For example, a 1.6 nm spacing corresponds to $g = 0.625$ nm$^{-1}$ in FFT graphs extracted from cryo-EM micrographs, while the X-ray scattering peak corresponding to this spacing appears at 3.925 nm$^{-1}$.

## Small-angle X-ray scattering (SAXS)
SAXS measurements were performed at the LiX (16-ID) beamline of the National Synchrotron Light Source NSLS-II, Brookhaven National Laboratory (New York, US). Two detectors, Pilaturs3X 1 M and Pilatuds3X 900 K, were placed with sample-to-detector distances of 3.566 m and 0.315 m, respectively. The combination of the two detectors covers a scattering range of $q$ from around 0.007 A$^{-1}$ to 2 A$^{-1}$. Sample solutions of BCP were prepared with a concentration of 50 mg mL$^{-1}$ in DI water. For TR-CP, 3.6 wt% samples in DI water were used, and loaded into an 8-cell flat window holder. All the samples were measured at constant temperature (20 °C, below LCST, and 40 °C, above LCST) with an acquisition time of 1 s per frame for a total of 20 frames on different spots of the sample cell. The scattering of the empty cells and cells filled with DI water was also measured to be used for background subtraction. Background subtraction and averaging of the data were performed using JupyterHub, which was provided by Brookhaven National Laboratory.

## X-ray photoelectron spectroscopy (XPS)
XPS was performed on Thermo Scientific K-Alpha XPS at the University of Delaware to determine the relative ratio of each polymer in the TR-CPs. Spectra of the samples were analyzed using Thermo Scientific Avantage Data System software. Five to six points on each thin film sample of the TR-CPs were taken to ensure random sampling of the heterogeneous film, and the data were averaged out. The sulfur peaks for PEDOT and PSS were observed at 164 eV and 168 eV, respectively, while the nitrogen peak for PNIPAM was observed at 400 eV.

## Raman spectroscopy
In-situ Raman measurements were recorded with a Jasco probe Raman spectrometer RMP-520 Series using a 785 nm immersion laser. Measurements were taken 20 s at a time, with 5 accumulations. Manual baseline and background corrections were applied to the spectra post-collection to minimize the background and baseline of water.

## In vitro cytotoxicity
In vitro cytotoxicity tests were performed on pure TR-CP gels based on the ISO 10993-5 protocol. We chose L929 cells because the cell line is recognized by the ISO 10993-5 standard and is widely accepted for cytotoxicity testing. L929 Fibroblasts were obtained from ATCC, derived from normal subcutaneous areolar and adipose tissue of a 100-day-old male C3H/An mouse. Cell line was tested for mycoplasma contamination before receipt by lab. For in vitro studies, $n = 3$ biological replicates were used per sample to account for inherent biological variability between different cell cultures. We initially prepared 3 control gels and 5 experimental gels. During handling, some of the gels

experienced mechanical damage (i.e., broke) before any outcome measurements could be collected and were therefore excluded as technical failures. The final analyzed sample sizes are reported in each figure legend.

## Indirect extract tests
Before performing cytotoxicity experiments, 200 μL TR-CP gels were added to a glass vial and incubated at 37 °C for 10 min for gelation. DMEM/F-12 media supplemented with both 10% fetal bovine serum (FBS) and 1% antibiotic–antimycotic (warm, 2 mL for 48-h test or 6 mL for 7-day test) was added carefully into the glass vial, soaking the TR-CP gels and incubated at 37 °C with 5% CO$_2$ in a humidified incubator for 24 h to create gel extracts from day 1 (D1 extracts). After collection of the complete DMEM/F-12 media, another 2 mL or 6 mL of warm media was added into the vial and incubated for 24 h to create gel extracts from day 2 (D2 extracts). The previous steps were repeated to create D3-D5 extracts. All the collected extracts were sterile-filtered with 0.22 μm membrane filters and stored at −80 °C for further study. Separately, a suspension of L929 cells was prepared at a concentration of $5 \times 10^4$ cell mL$^{-1}$ and 100 μL of the suspension was dispensed into each well of a 96-well plate. The plate was incubated at 37 °C with 5% CO$_2$ in a humidified incubator. After 24 h, culture media were removed from the wells and replaced by 100 μL of fresh complete media (Control) or extract media of the TR-CP gels from day 1 to day 5 (D1-D5 extracts). Following 48 h or 7 days, the cell viability was assessed via alamarBlue™ assay (Invitrogen; Cat no. A50101) and read by Cytation3 plate reader (BioTek). Samples were repeated in triplicate.

## Direct contact tests
TR-CP gels were lyophilized and sterilized using ethylene oxide (ETO) sterilization (Anprolene AN75, Anderssen) and allowed to rest for 24 h. The sterilized TR-CP gels were reconstituted with complete DMEM/F-12 media or with 1% w/v sodium alginate solution, giving a final concentration of 3.6% or 5.6% w/v. L929 cells were prepared at a concentration of $5 \times 10^5$ cell mL$^{-1}$ and 10 μL of this suspension was mixed with 90 μL of reconstituted TR-CP gels, then added into a 35 mm glass bottom petri dish. 2 mg/mL collagen gel (First Link (UK) Ltd.) or 1% w/v sodium alginate (Sigma-Aldrich) gel was used as control gel. A commercially available polycarbonate membrane (Sterlitech PCTF 0425100) with 0.4 μm pore size was attached to the top of the well to protect the control or TR-CP gel. The petri dishes were incubated at 37 °C with 5% CO$_2$ in a humidified incubator for 10 min for gelation of the pure TR-CP gels, or a crosslinking solution (20 mM barium chloride, 5% mannitol) was used for the gelation of the TR-CP/alginate gels. Upon gelation, 3 mL of warm complete DMEM/F-12 media was carefully added to each dish. Samples were repeated in triplicate. Following 48 h, a live/dead solution was prepared from the LIVE/DEAD® Viability/Cytotoxicity Kit (Invitrogen; Cat no. L3224) with 0.5 μL of calcein AM and 2.0 μL of ethidium homodimer-1 (EthD-1) in 1 mL of D-PBS. Each well of gels was stained with live/dead solution (100 μL), incubated for 30 min, and imaged with a Nikon W1 Dual CAM spinning disc confocal laser microscope. The cell viability percentage was calculated using Fiji (ImageJ).

## In vivo cytocompatibility
**Subcutaneous implantation of TR-CP/Alginate composite gels.** Male Sprague Dawley rats (3 months old) were ordered from Charles River Laboratories. All animal protocols were approved by Northwestern University's Institutional Animal Care and Use Committee with approval number IS00024034. Rodents were housed in conventional rat housing. All animals were housed under a 12-h light/12-h dark cycle with ad libitum access to standard chow and water at the Center for Comparative Medicine, Northwestern University (Chicago, IL). 2–5% Isoflurane was used as anesthetic. For pain management, Meloxicam (1–2 mg/kg, SC q12-24h) was administered immediately prior to bioink implantation. 12–24 h of the first dose, the animal received a second

dose of Meloxicam. Two animals were used, 1 per time point, with $n = 2$ biological replicates per animal. Collected data was merely qualitative, thus a small sample size was sufficient.

The in vivo performance of the gels was evaluated by subcutaneous implantation. To minimize animal impact, animal sample size was $n = 2$, each animal had $n = 4$ samples implanted for $n = 4$ per condition. Four 1 cm incisions were made through the skin into the subcutaneous space to create a pocket to accommodate a sterile 0.7 cm diameter, 0.5 cm thickness gel. Implants were placed into the pockets, and the wound was closed with wound staples. One rat received four implants at both sides of the back, two control alginate gels, and two PEDOT-alginate gels. Gels were explanted at 1- and 2-weeks post implantation after rats were euthanized. The previous incision sites were identified, and the subcutaneous pockets were re-expanded. The gels were located, and skin samples were collected containing the gels.

**Tissue processing.** The collected skin samples were fixed using 4% paraformaldehyde overnight and washed with PBS 3x. After fixation, the tissues were dehydrated in ethanol solutions (70%, 90%, and 100% twice) for 45 min each. The tissues were then cleared in successive xylene solutions for 1 h twice, and placed into paraffin overnight at 60 °C. The tissues were then embedded in a paraffin mold by the Mouse Histology & Phenotyping Laboratory (MHPL) at Northwestern University.

**Histological and immunohistochemical analysis.** The tissues were sectioned at 5-μm thickness onto microscope slides. Tissue sections were then stained for hematoxylin and eosin (H&E) and imaged under a brightfield microscope.

### Confocal microscopy
A Nikon W1 Dual CAM spinning disc confocal laser microscope was used to record images of cells for the live/dead assays.

### Self-healing of the TR-CP
To observe self-healing in TR-CPs, 0.3 mL of 5.6 wt% TR-CP liquid was drop-casted on a clean glass slide, pre-heated to 37 °C. Once the TR-CP was gelled, it was sliced with a razor blade into two halves and separated. Then, the two halves were manually joined, and the gel was allowed to cool slightly to self-heal. Finally, the glass slide was heated again to re-form the gel.

### Injection of the TR-CP into a viscous liquid or gel
To demonstrate the injectability of TR-CPs, a viscous liquid and a hydrogel substrate were used. For the viscous liquid, 1.8 wt% solution of sodium alginate in water was prepared. The solution was pre-heated to 37 °C. Then, 5.6 wt% TR-CP, dispersed in PBS, was injected into the warm alginate solution using a blunt 21 G needle to form a continuous TR-CP "wire". The wire remained stable as long as the temperature was maintained above 37 °C, displaying no spreading once the injection was complete (Fig. 6b) and at least for 20 days. The dispersion was also injectable through a 30 G needle. On cooling the alginate solution to room temperature, the TR-CP cooled down to a liquid and spread (Fig. S32).

### Surface electromyography (s-EMG) measurements
**Device design.** For sEMG, a custom set-up was designed. Two hollow PDMS reservoirs were prepared and Ag/AgCl button electrodes were removed from commercially available 3M™ Ag/AgCl Red Dot Monitoring Electrodes. Copper wire was attached to the metal side of the button electrodes, and the connection was sealed with copper tape. This electrode set-up was placed within the reservoir, and 5.6 wt% TR-CP liquid (in PBS) was poured into it, ensuring good coverage on the electrode. The TR-CP/electrode assemblies were pre-heated to 40 °C

until the TR-CPs gelled, and then attached to the forearm. A commercially available transparent dressing, 3 M Tegaderm™ was used to cover and seal the assembly on the skin. Finally, a commercially available heating patch (CVS Health™ Heat Therapy Patch) was applied on top to maintain a constant temperature during the measurement.

**Measurements.** The s-EMG measurements were performed using the methods described by Blau et al.[43]. The electrodes were connected to the ECG channel of a MAX30001- EVSYS evaluation board (Maxim Integrated Products, Inc.) using alligator clamp wires. The reference electrode (3M™ Ag/AgCl Red Dot Monitoring Electrode) was placed on the elbow of the same arm of a volunteer and connected to the Body Bias pin of the board. The participant in Fig. 6 provided written, informed consent. The ECG channel was set to a gain of 20 V/V and a sampling rate of 512 readings s$^{-1}$. Post-ADC digital filters were used during the recordings. The low pass filter cutoff was 40 Hz, and the high pass filter cutoff was 0.5 Hz. A Butterworth notch filter with a width of 59 to 61 Hz was applied post-measurement to eliminate 60 Hz power-line noise. Fist opening and closing movements were performed every 5 s to record changes in signal amplitude on movement.

### Reporting summary
Further information on research design is available in the Nature Portfolio Reporting Summary linked to this article.

## Data availability
The source data for NMR, GPC, rheology, EIS, SAXS, XPS, Raman spectroscopy generated in this study have been deposited in the Figshare database under accession code https://doi.org/10.6084/m9.figshare.29260157. All other data generated and analyzed in this study are provided in the Supplementary Information file: synthesis and materials preparation, molecular characterization: NMR and GPC for PSS CTAs and PSS-b-PNIPAM, Experimental design, rheological characterization of TR-CP, effect of TR-CP concentration. Stability of TR-CP, electronic characterization of TR-CP, TR-CP microstructure characterization using cryo-EM, rheology, SAXS, XPS, and Raman spectroscopy, electronic characterization of TR-CP at pH = 7, in vitro cytocompatibility, characterization of TR-CP/alginate, in vivo cytocompatibility, and subcutaneous implantation of TR-CP/alginate gels, processing of TR-CP, and surface EMG measurements. Data is available upon request to the corresponding author L.V.K.

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

## Acknowledgements

This work was supported by a National Science Foundation (NSF) CAREER award (grant No. DMR-2237888) (synthesis and characterization), a Beckman Young Investigator award from the Arnold and Mabel Beckman Foundation (dx.doi.org/10.13039/100000997) (processing and applications), and a University of Delaware Research Foundation (UDRF) seed funding to L.V.K. X.X., and J.R. acknowledge support from the Army Research Office under Cooperative Agreement Number W911NF-23-2-0138. The views and conclusions contained in this document are those of the authors and should not be interpreted as representing the official policies, either expressed or implied, of the Army Research Office or the U.S. Government. The U.S. Government is authorized to reproduce and distribute reprints for Government purposes, notwithstanding any copyright notation herein. R.D. and J.R. acknowledge support from NIH grant 5T32EB031527-04. K.S. acknowledges funding support from National Institute of Standards and Technology (NIST), Department of Commerce under agreement #370NANB17H302. W.X. and D.P. acknowledge support by NSF through the University of Delaware Materials Research Science and Engineering Center (MRSEC) (DMR-2011824). M.G. and E.D.G. acknowledge support from NSF under Award DMR-1905550. C.L.C. acknowledges support from NIH grant 5T32HL094293-14. The use of facilities and instrumentation at the University of Delaware was supported by the National Institutes of Health (NIH), NSF awards CHE-0421224 (NMR), and CHE-1428149 (XPS). SAXS experiments (W. X.) were performed at the LiX beamline of the National Synchrotron Light Source II, a U.S. Department of Energy (DOE) Office of Science User Facility operated for the DOE Office of Science by Brookhaven National Laboratory under Contract No. DE-SC0012704. The LiX beamline is

part of the Center for BioMolecular Structure (CBMS) which is primarily supported by the NIH NIGMS through a Center Core P30 Grant (P30GM133893), and by the DOE Office of Biological and Environmental Research (KP1607011). K. O. acknowledges support from the University of Delaware Startup funds. The authors would like to thank Prof. Norman Wagner for access to the AR-G2 rheometer, Prof. Emil Hernandez-Pagan for access to the in-situ Raman spectrometer, and Prof. David Martin for access to electronic characterization equipment. We acknowledge Dr. Sung Hyun (Joseph) Cho's assistance with cryo-EM data acquisition and the cryo-EM facility available in Huck Institutes of the Life Sciences at Penn State University. We would also like to thank Tulika Bhattacharya for her help in conducting rheology experiments, and Yaping Wang and Yong Zhao for their help in collecting preliminary data for TEM and cryo-EM. National Science Foundation, DMR-2237888: L.V.K. Beckman Young Investigator award: dx.doi.org/10.13039/100000997: L.V.K. University of Delaware Research Foundation: L.V.K. Army Research Office W911NF-23-2-0138: X.X. and J.R. National Institutes of Health: 5T32EB031527-04: R.D. and J. R. National Institute of Standards and Technology (NIST), Department of Commerce #370NANB17H302: K.S. University of Delaware Materials Research Science and Engineering Center (MRSEC) (DMR-2011824): W.X. and D.P. National Science Foundation, DMR-1905550: M.G., E.D.G. National Institutes of Health 5T32HL094293-14: C.L.C. National Institutes of Health (NIH), NSF awards CHE-0421224 (NMR), and CHE-1428149 (XPS): V.S.D., C.L., J.A.A. N.Y., T.S., and L.V.K.

## Author contributions

V.S.D. and L.V.K. conceived the project and designed the experiments. V.S.D. synthesized the materials. X.X. performed in vitro cytocompatibility experiments and analyzed the data. R.D. and C.L.C. performed in vivo biocompatibility experiments and analyzed the data. V.S.D. and K.S. designed and performed rheology experiments and analyzed the data. M.G. performed the cryo-EM experiments, and M.G. and E.D.G. analyzed the data. W.X. performed the SAXS experiments. R.W. modelled and analyzed the SAXS data. V.S.D., Y.W., and C.Y.L. performed electronics measurements and analysis. J.A., C.L., and A.N.Y. assisted with the materials synthesis. T.S. performed XPS experiments and helped V. S. D. with data analysis. K.O. and V.S.D. performed in-situ Raman experiments and analyzed the data. V.S.D. designed and developed the s-EMG electrodes. D.M.N. and V.S.D. performed the s-EMG experiments and analyzed the data. D.P., E.D.G., J. R., and L.V.K. provided guidance and funding. All authors contributed to writing and editing the manuscript, and have approved the final version of the manuscript.

## Competing interests

The authors declare the following competing interest: V.S.D. and L.V.K. are inventors on a patent application filed by the University of Delaware on the "Thermo-responsive and conducting polymer with a reversible sol gel transition" (PCT International Patent Application No. PCT/US2023/015527, patent pending). The synthesis of the TR-CP and preliminary rheological and electrochemical characterization are included in the patent application. The remaining authors declare no competing interests.
