## [Transparent Peer Review file · Nature Communications]

Thermo-reversible gelation of self-assembled conducting polymer colloids

Corresponding Author: Dr Laure Kayser

This file contains all reviewer reports in order by version, followed by all author rebuttals in order by version. Parts of this Peer Review File have been redacted as indicated to maintain the confidentiality of unpublished data.

Version 0:

Reviewer comments:

Reviewer #1

(Remarks to the Author)

Recommendation: Revisions needed as noted.

Suitable for publication after revisions.

The manuscript by Kayser and team entitled “Thermo-reversible gelation of self-assembled conducting polymer colloids” presents an advancement in the development of thermo-reversible conducting hydrogels (TR-CPs) with reversible sol-gel transitions for various bioelectronic applications, notably for in vivo. The novelty of their work compared to similar ones in literature (<https://www.nature.com/articles/s41428-022-00626-y>, <https://onlinelibrary.wiley.com/doi/full/10.1002/adfm.202403708>) lies in the versatile synthesis method, combining PEDOT:PSS and PNIPAM via block copolymer synthesis. Their approach shows clear advantages over irreversible covalent crosslinking methods and synthesis design criteria on achieving desirable sol gel transitions. The authors show in vitro and in vivo cytocompatibility as well as wearable electrophysiology sensing via electromyography electrodes and highlight (but do not showcase) their potential use for injectable hydrogel bioelectronics (in vivo).

The manuscript is well-written, experimental design is robust, employing a suite of comprehensive characterization techniques such as rheology, cryo-EM, SAXS, and electrochemical impedance spectroscopy to provide an in-depth understanding of the material's behavior properties and mechanism. Control experiments comparing block and random copolymer architectures highlight the critical role of block copolymers in achieving the desired gelation properties. The results are clearly presented, and the data (figures and tables) largely support the conclusions. I believe this work is suitable for publication in Nat Comms, however, several areas require clarification or additional data:

1. Line 118–121: The manuscript states that TR-CP gels are stable for at least 15 minutes at 37°C and several months at room temperature in closed vials. Are the gels stable at physiological temperatures (37°C) for longer durations?
2. Line 256–258: Characterization of TR-CP gels reconstituted in cell media is noted. However, discuss how basic pH affects the gels and its implications for bioelectronic applications, especially on skin where pH may vary.
3. Line 330: The manuscript mentions self-healing behavior of gels when cooled and reheated to 37°C. Specify the time required for self-healing.
4. Figure 4: Why in Fig 4b fibroblasts appear to have adopted their typical spindle-like structure while in all other cases we do not observe this? Please comment
5. Figure 6: :s-EMG electrode: Please comment on the following:
 - versatility and re-usability is claimed however in terms of versatility I do not see practical implementation of the electrode if continuous heat is needed? Hence for continuous on body sensing this electrode does not seem to be practical. Please add a small discussion addressing this
 - Self-adhesiveness: irritation is mentioned as an additional advantage of this electrode vs commercial however irritation comes also from the adhesives used in wearable electrodes and this electrode does not seem to be self-adhesive. Please add a small discussion addressing this
6. Gel dissolution: Elaborate on the mechanisms of gel dissolution under physiological conditions (e.g., enzymatic activity or ionic strength).
7. In vivo cytocompatibility (Section 3.11.1): Specify sample size and statistical methods (no of animals) used in these studies.
8. Figure S3: Provide clearer interpretation of temperature-dependent rheological behavior, specifically trends in G' and G'' and their relation to molecular dynamics.

9. Electronic characterization:

-Discuss the physical implications of the changes in the circuit elements (e.g., R_c , R_e , and R_i) reported in Tables S1 and S2.
- Why Debye and not Randle circuit modelling? (given the electrically conducting properties of the hydrogel). Please include more discussion on the fitting that was used and the underlying hypothesized mechanisms of conductivity within this hydrogel.

10. Cryo-EM methodology (Section 1.7): Address potential artifacts arising from sample preparation.

11. Units and symbols: Standardize units across sections (e.g., mS/cm vs. $\text{mS}\cdot\text{cm}^{-1}$).

12. In vivo biocompatibility: Is there any in vivo degradation analysis over time?

13. Control experiments: Is there any control with gels formed with PNIPAM alone and covalently crosslinked PEDOT:PSS ? to validate the superiority of TR-CPs.

14. Figure captions: Enhance captions with details. For instance:

o Fig. 2: Explain how PEDOT loading affects ionic conductivity.

o Fig. 5: Describe observed microvascularization in histological images.

o Table 1: Justify why the specific mass ratio of PSS:PNIPAM selected for gelation properties.

15. Repeated exposure: Discuss whether repeated thermal cycling of gels could lead to cellular or tissue damage, particularly due to material degradation or residual chemicals.

16. Typos: Line 271 states, "he TR-CPs were then incorporated..." This appears to be a typo and should be corrected to "the TR-CPs."

These revisions, are sought to strengthen the manuscript's impact and make it further suitable for publication.

Reviewer #2

(Remarks to the Author)

Damani et al. did a systematic study of the polymer hybrid, covering from design through synthesis and characterization to their applications in bioelectronics. The story is very interesting and should interest broad readers in the fields of chemistry, materials science and bioelectronics. The following points should be addressed in the revision, to improve the clarity of the story.

1. One major concern is the structure information of the PEDOT:PSS-PNIPAM, based on SAXS and Cryo-EM. In general, it seems that they are not consistent with each other. Those nanostructures like lamellae or particles, as mentioned in page 8 (lamellar fringes with ~ 1.6 nm spacing and small particles (~ 5 nm)) are not observed in the SAXS data. Although the SAXS data fitting based on Guinier-Porod model could provide some structure information. Technically, the fitting procedure is blurred by the confusing descriptions: on one side, the authors mentioned the reference measurement and background correction in the SAXS experiment section; on the other side, "Two peaks (Figure S13) in the high q region ($q = 15 \text{ nm}^{-1}$ and $q = 28 \text{ nm}^{-1}$) were attributed to the sample holder cells and water background, and excluded from the fitting.", which shows clearly these two peaks were not subtracted for the fitting (Figure S13 of SI). Meanwhile, how were the two Guinier-Porod models used for the data fitting? Is this method reliable for such a complex system? Correlations with the cryo-EM data plays vital role.

2. It is recommended that the authors should include more evidences to support their judgements like "These lamellar fringes are likely associated with a semi-crystalline PEDOT phase. The small size of these particles in solution likely explains the excellent colloidal stability of the dispersion and their ability to be lyophilized and redispersed.". If the PEDOT was crystallized, which should also be observed with the SAXS data? Did the authors cross-check and exclude the artefacts caused by ice formation, for the cryo-EM data?

3. It's a bit confusing about the description of the colloidal behavior of the PEDOT:PSS-PNIPAM. If the authors believe it was a colloid, then please provide the structure information such as size and shape. In addition, it is important to illustrate the dominating factors for their stable dispersion in the solvent. It should be helpful to understand the structure-property correlations if the authors could clarify the colloidal and hydrogel behavior of the PEDOT:PSS-PNIPAM.

4. It is not very clear why the authors include a series of PSS-b-PNIPAM samples. The logic connection with the main story should be clarified in the main text.

5. Did the authors remove the residue of the catalysis and oxidant during the synthesis of the PEDOT:PSS-PNIPAM. In principle, $\text{FeCl}_3/\text{Na}_2\text{S}_2\text{O}_8$ and their derivatives would influence the electronic and ionic properties of the sample. It could also bring challenges for the conductivity measurements since, the copper used in electrical conductivity (e.g., Figure S6) measurement might react with FeCl_3 , for example. In addition, these residues might bring negative influence for their bioelectronic applications. Please clarify these points.

6. It should be helpful if the authors could illustrate the dominating factors for the electronic and ionic conductivities of the PEDOT:PSS-PNIPAM. Based on which, discussions about the changes of the conductivities as a result of temperature change, in the section of "Electronic properties of the TR-CP". In addition, correlations with the microstructures mentioned in the following section should help the readers to understand the story better.

7. It would be very helpful if the authors could include more related references, especially in the results and discussion part.

8. Line 186, P7, the authors wrote that "To elucidate the nanoscale structure of the TR-CP and what prompts it to form a porous network,". What is the porous network mean?

9. It should be a mistake to use the terms like molar ratio of PSS:PEDOT molar ratio. In principle, the authors could get the numbers of SS units and the EDOT that used for the synthesis. Meanwhile, do the authors have the M_w information of PEDOT?

10. Please clarify the sentence in the SI, "We chose 37°C for solution temperature as it allows the formation of vitreous ice layers which are thin enough for TEM imaging while maintaining the gelation behavior of the block copolymer."

Reviewer #3

(Remarks to the Author)

This manuscript reports a new conductive thermos responsive hydrogel with reversible gelation based on a PEDOT:PSS poly(N-isopropylacrylamide) (PSS-b-PNIPAM) complex. The reversible gelation occurs at 35 degrees Celsius. Overall, the manuscript gives a very detailed description of the synthesis and the characterization of the hydrogel. However, its relevance for the material science community is not very well substantiated. As I do not see any potential breakthrough in terms of materials, I suggest rejection and publication in a more specialized journal.

Please find additional comments below.

1. By which mechanism does the impedance reduction of TR-CP occur due to temperature changes and consequently upon gelation? According to the numbers given in page 5 and table S1, the effect on the temperature on electronic and ionic resistance is comparable, with electronic resistance decreasing by about 50% and ionic resistance increasing by about 30%. It is unclear why the effect on ionic conductivity is not discussed. Also, in the tables S1-S3 the values are given with several significant digits without error bars.
2. The lowest impedance was observed when the PSS:PEDOT ratio was 3.5:1, how can the conduction mechanism of PEDOT:PSS be explained, considering the difference between ionic conductivity and electronic conductivity?
3. The ionic conductivity refers to which ion?
4. Figure 2d: is the circuit closed only upon gelation? 2f: the caption says: "Nyquist plot for the TR-CP at 5.6 wt% in PBS (pH = 4) which showed the highest conductivity." It is unclear what shows the highest conductivity.
5. Why did the conductivity decrease when the concentration increased to 7.6 wt%? Can you explain in more detail?
6. The conductivity calculation results of 5.6wt% TR-CP ($\sigma_i = 200 \text{ mS cm}^{-1}$, $\sigma_e = 14 \text{ mS cm}^{-1}$) was mentioned to be similar to those of existing conductive hydrogels. However, did this study specify which specific conductive hydrogel was compared?
7. Why did the authors choose a two-point probe method instead of a four-point probe for measuring conductivity in the TR-CP samples?
8. The concentration of 3.6 wt% was determined to be the lower concentration for gel formation. Could authors explain what changes occurred in the microstructure of TR-CP as the concentration increased?
9. The statement "...an increased inflammatory response is common when conducting polymers are implanted" should be used with caution. This is supported only by two quite old citations, while many other studies do not show similar evidence.
10. For the application point of view, the advantage of the material remains unclear both for invasive and non-invasive bioelectronics. The EMG epidermal electrodes do not seem to have a particular advantage with respect to similar materials based on PEDOT which are also easier to fabricate.
11. The relevance of self-healing properties is unclear. Also, it is unclear the time required for the process and how mechanical and electrical properties change before and after healing.
12. In terms of cytocompatibility, the systems behave similarly to other hydrogels and electrode materials based on PEDOT:PSS. Also explain why that type of cell was selected.

Version 1:

Reviewer comments:

Reviewer #1

(Remarks to the Author)

The authors adequately reviewed the manuscript which improved significantly. I recommend its publication without further revisions.

In particular, all my comments have been addressed with additional experiments and explanation in the text when needed substantially improving clarity and strengthening their data to better support the conclusions.

With respect to reviewer 3, all the comments have been adequately addressed as well, including more clarification on the

methods for assessing electrical/ electrochemical performance of the gels, conductivity calculations, elaboration on the structural changes at different concentrations, the gel healing time and finally cytocompatibility tests. Overall the authors have performed a very detailed revision with additional experiments, references to support the claims and more discussion in the text better clarifying the methodology, results and further supporting the conclusions.

Reviewer #2

(Remarks to the Author)

While the authors have made an effort to address the concerns raised in the initial round of reviews, several critical issues—particularly regarding data reliability and consistency across techniques—remain inadequately resolved. These deficiencies significantly compromise the validity of the structural and mechanistic claims made in the manuscript. I outline the major points below:

1. FFT and Cryo-EM Data Interpretation Remains Ambiguous:

- a. The azimuthally integrated FFT data in Fig. S13g show negative "n" values and poorly resolved features. It is unclear how the spatial frequency data are derived or how peak assignments (e.g., 1.57–1.65 nm) are extracted.
- b. The authors might incorrectly reference a "missing" 6 nm^{-1} signal, even though the maximum q in their FFT only reaches $\sim 3 \text{ nm}^{-1}$. The claimed absence of a signal cannot be used to conclude structural nonexistence without extending the q-range to at least 10 nm^{-1} .
- c. The connection between 6 nm^{-1} and 1.3 nm in the caption of Fig. S13 is not justified or physically explained.
- d. In Fig. S13h, the real-space peak-to-peak distance decreases as q increases, which is contradictory to the authors' claim of invariant lamellar spacing. This trend implies dynamic changes in local structure or defocus artefacts, both of which need detailed clarification.
- e. It remains unclear why the FFT-derived azimuthal profiles (panel g) are used to make statements about the absence of PEDOT stacking. The logic of assigning signal absence based on azimuthal—not radial—FFT profiles is questionable.
- f. Furthermore, there is a notable inconsistency between panel g and panel h of Fig. S13. The data sources and processing parameters should be disclosed in more detail to explain this discrepancy.

2. SAXS Data Interpretation Is Weak and Inconsistent with Cryo-EM:

- a. The SAXS data in Fig. R3 show clear peaks at $q = 5.6$ and 15 nm^{-1} . These peaks are only briefly addressed and ambiguously attributed to PNIPAM interactions or background, without proper experimental validation.
- b. The authors' claim that the peaks at $q = 15$ and 28 nm^{-1} are background-related must be supported with separate background-subtracted plots or appropriate references. Without this, their exclusion from fitting is speculative and methodologically unsound.
- c. Discrepancy in the q-range for SAXS measurements at different temperatures (e.g., $20 \text{ }^\circ\text{C}$ vs $40 \text{ }^\circ\text{C}$) raises serious questions about data comparability. The authors provide no explanation for this inconsistency.

3. Cryo-EM Structural Claims Are Physically Incongruent:

- a. The claim that lamellar fringes seen in Cryo-EM correspond to crystalline PEDOT is not reconciled with SAXS/WAXS data, where no corresponding features are found—even though PEDOT crystalline domains ($\sim 6 \text{ nm}$) should be visible in high-q scattering, as shown by prior literature (e.g., Chem. Mater. 2016, 28, 9, 3185–3192).
- b. The core-shell particle model ($\sim 5 \text{ nm}$ radius) is inconsistent with the $\sim 6 \text{ nm}$ crystalline domains of PEDOT. If PEDOT were crystalline and formed the particle core, one would expect scattering or diffraction consistent with this structural scale.
- c. It is conceptually inconsistent that PEDOT is visible in Cryo-EM while the more abundant and chemically similar PSS and PNIPAM components are not. The authors must clarify the contrast mechanism, phase contrast vs Z-contrast, and consider charge density and hydration shell factors.

4. Possible overinterpretation and Misrepresentation of SAXS/WAXS Models:

- a. The use of a core-shell sphere model oversimplifies the likely highly anisotropic, lamellar, or cylindrical features observed in Cryo-EM. The lamellar fringes ($\sim 20 \text{ nm}$ long, $\sim 5 \text{ nm}$ wide) are inconsistent with spherical scattering models.
- b. The core radius of 3.2 nm (inferred from SAXS) contradicts literature reports ($\sim 30 \text{ nm}$) and the authors' own assertion that PEDOT forms $\sim 6 \text{ nm}$ crystals. This discrepancy is not addressed.
- c. The authors make unsubstantiated claims about lamellar features being "too large to be π - π stacking," while lamellar π -stacking is routinely observed at similar dimensions in PEDOT-rich phases in aqueous dispersions via WAXS.

Given the inconsistencies between techniques, questionable data interpretation, and unresolved structural ambiguities, I do not believe the manuscript in its current form meets the standards required for publication in Nature Communications. A complete revision with improved experimental methodology, transparent data processing, and consistent structure-property correlation is required before reconsideration.

Version 2:

Reviewer comments:

Reviewer #2

(Remarks to the Author)

[Note from the Editor: Reviewer 2 left comments for the editor only]

Reviewer #4

(Remarks to the Author)

[Note from the Editor: Reviewer #4 assessed the response given to reviewer #2]

Review: "Thermo-reversible gelation of self-assembled conducting polymer colloids"

This work explores a thermo-responsive conducting polymer that undergoes a fully reversible non-covalent crosslinking at 35 °C within less than a minute to form conductive hydrogels. The thermo-responsive conducting polymer is based on a PEDOT:PSS-b-PNIPAM block copolymer. A material with outstanding performance.

The manuscript in the revised form has been significantly improved; however, there are still several issues that need to be clarified before its publication:

1. In the caption of Figure S12, the meaning of values 53 and 57 nm should be explained.
2. In the caption of Figure S13, should be indicated if it is either below or above LCST.
3. The FFTs from micrographs a), b), and c) in Figure S13 should be included, and indicated how the azimuthal integration has been performed.
4. The term "azimuthal integration" instead of "radial integration" is extensively used for the X-ray scattering community in order to derive the $I(q)$ profiles from 2D patterns.
5. Concerning Figure S16d) and e), what is the background subtracted to the raw data? The authors could try to subtract the signals from the sample holder and water.
6. What is the meaning of the Gaussian Peak 1 (centered at about 0.1 nm⁻¹) in Figure S16c)?

Reviewer #1 (Remarks to the Author):

Recommendation: Revisions needed as noted.

Suitable for publication after revisions.

The manuscript by Kayser and team entitled “Thermo-reversible gelation of self-assembled conducting polymer colloids” presents an advancement in the development of thermo-reversible conducting hydrogels (TR-CPs) with reversible sol-gel transitions for various bioelectronic applications, notably for in vivo. The novelty of their work compared to similar ones in literature (<https://www.nature.com/articles/s41428-022-00626-y>, <https://onlinelibrary.wiley.com/doi/full/10.1002/adfm.202403708>) lies in the versatile synthesis method, combining PEDOT:PSS and PNIPAM via block copolymer synthesis. Their approach shows clear advantages over irreversible covalent crosslinking methods and synthesis design criteria on achieving desirable sol gel transitions. The authors show in vitro and in vivo cytocompatibility as well as wearable electrophysiology sensing via electromyography electrodes and highlight (but do not showcase) their potential use for injectable hydrogel bioelectronics (in vivo).

The manuscript is well-written, experimental design is robust, employing a suite of comprehensive characterization techniques such as rheology, cryo-EM, SAXS, and electrochemical impedance spectroscopy to provide an in-depth understanding of the material's behavior properties and mechanism. Control experiments comparing block and random copolymer architectures highlight the critical role of block copolymers in achieving the desired gelation properties. The results are clearly presented, and the data (figures and tables) largely support the conclusions. I believe this work is suitable for publication in Nat Comms, however, several areas require clarification or additional data:

Our response: We thank Reviewer #1 for reviewing our manuscript, and recognizing the novelty of our work and our comprehensive materials characterization efforts. We appreciate your insightful comments, which will indeed improve the quality of the manuscript. We have addressed them point-by-point below.

1. Line 118–121: The manuscript states that TR-CP gels are stable for at least 15 minutes at 37°C and several months at room temperature in closed vials. Are the gels stable at physiological temperatures (37°C) for longer durations?

Our response: We performed an additional experiment to test the stability of the gel over longer times. We formed the TR-CP gel in a vial by placing and maintained it at 37 °C for 80 days. No changes were observed during this extended time.

Our changes to the manuscript: The following text was added to the manuscript:

“To further test the stability of the TR-CP at 37 °C, the TR-CP gel was formed inside a sealed vial, and maintained at 37 °C for 80 days. Under these conditions, the TR-CP remained gelled with negligible changes in volume (**Fig. S6**). After 80 days, the TR-CP still retained its reversible sol-gel transition, showcasing the robust and stable nature of the gel.”

And we have added the following figure to the SI:

Fig. S6. Stability of the TR-CP gel at 37 °C for 80 days. ~ 0.5 mL of the 5.6 wt% TR-CP in DI water was added to a vial, and heated to 37 °C in an oil bath. The temperature of the bath was maintained at 37 °C for 80 days. **a.** Picture of the TR-CP gel on Day 1 **b.** Picture of the TR-CP gel on Day 80, showing negligible changes in volume and stability. **c.** Picture of the TR-CP reversed to its liquid state on cooling to room temperature (23 °C), after 80 days of heating at 37 °C.

2. Line 256–258: Characterization of TR-CP gels reconstituted in cell media is noted. However, discuss how basic pH affects the gels and its implications for bioelectronic applications, especially on skin where pH may vary.

Our response: We performed additional experiments to test the TR-CP at basic pH. We note that a discussion on acidic vs neutral pH was already included in the original manuscript. The lyophilized TR-CP was reconstituted in PBS, and the pH was adjusted up to pH = 14 using aqueous NaOH. We observed that the TR-CP retained its reversible sol-gel transition up to pH = 12; however, at that pH the color of the TR-CP changed from blue-green to a blue-violet color, which could be indicative of the de-doping of PEDOT.

Our changes to the manuscript: The following text has been added to the manuscript for clarification of the effects of pH on the gels:

“The gels can withstand a range of pH conditions and retain their sol-gel transition below 37 °C even under basic conditions, up to pH = 12 (**Fig. S21**), highlighting that this material can be used under a range of different physiological conditions.”

And have added the following figure and text to the SI:

Methods:

“Preparation of the TR-CP solution at basic pH. PEDOT:PSS₉₆-*b*-PNIPAM₄₄₀ (TR-CP) was lyophilized and reconstituted in 1X PBS (5.6 wt%). 0.1M NaOH was added dropwise using a micropipette to adjust the pH to 9 and 12 without drastically changing the concentration. Additionally, to make a sample with pH = 14, 56 mg of the TR-CP was added to 1mL of 1M NaOH, and we attempted to disperse the TR-CP by stirring overnight.”

Fig. S21. Effect of basic pH on the TR-CP at pH = 9 (top) and pH = 12 (bottom). The lyophilized TR-CP powder was dispersed in PBS at 5.6 wt%, and 0.1M NaOH was added dropwise using a micropipette to adjust the pH. The TR-CP showed good dispersibility under both conditions and retained its reversible sol-gel transition. However, the TR-CP powder was not dispersible in 1M NaOH, indicating that pH = 14 was too harsh for the TR-CP.

3. Line 330: The manuscript mentions self-healing behavior of gels when cooled and reheated to 37°C. Specify the time required for self-healing.

Our response: The gels took 25 – 30s to heal for a volume of 0.6 – 1mL.

Our changes to the manuscript: The following text has been added to the caption of **Fig. 6a**:

“The gels took ~25 s to heal for a volume of 0.6 mL.”

4. Figure 4: Why in Fig 4b fibroblasts appear to have adopted their typical spindle-like structure while in all other cases we do not observe this? Please comment.

Our response: The hydrogel stiffness significantly impacts the morphology of L929 cells (**Fig. R1**). For the pure TR-CP gels, we chose 2 mg/mL of collagen gel as the control. As shown by Montalbano et. al.¹, the collagen gel is relatively stiff, so the L929 cells have enough adhesion sites to spread themselves and form an elongated spindle-like shape. Instead, the pure TR-CP gels, the alginate control gel, and the TR-CP/alginate gel have a much lower stiffness and lack the necessary protein-like sequences promoting mammalian cell attachment, which results in cells adopting a more rounded shape. An image of live/dead results for L929 cells in alginate gel from another paper by Klippel et. al.² is shown below to illustrate the rounded shape in soft gels like alginate and the TR-CP.

[REDACTED]

L929 in collagen, spindle-like shape¹

L929 in alginate, rounded shape²

Fig. R1: Cell morphology of L929 cells in (left) collagen; (right) alginate.

Our changes to the manuscript: The following text was added to the manuscript:

“In the collagen control group, the cells spread and adopted spindle-like shapes (**Fig. 4b**), while in the 3.6 wt% TR-CP and the TR-CP/alginate composite the cells exhibited less spreading and had a rounded shape (**Fig. 4c**). These differences are consistent with previous reports, and explained by the fact that the collagen is stiffer collagen than the TR-CP and TR-CP/alginate.”

The following references were added to this section:

1. Montalbano, G.; Toumpaniari, S.; Popov, A.; Duan, P.; Chen, J.; Dalgarno, K.; Scott, W. E.; Ferreira, A. M. Synthesis of bioinspired collagen/alginate/fibrin based hydrogels for soft tissue engineering. *Materials Science and Engineering: C* **2018**, *91*, 236-246. DOI: <https://doi.org/10.1016/j.msec.2018.04.101>.
2. Klippel, S.; Döpfert, J.; Jayapaul, J.; Kunth, M.; Rossella, F.; Schnurr, M.; Witte, C.; Freund, C.; Schröder, L. Cell Tracking with Caged Xenon: Using Cryptophanes as MRI Reporters upon Cellular Internalization. *Angew. Chem. Int. Ed.* **2014**, *53* (2), 493-496. DOI: <https://doi.org/10.1002/anie.201307290>.

5. Figure 6: :s-EMG electrode: Please comment on the following:

-versatility and re-usability is claimed however in terms of versatility I do not see practical implementation of the electrode if continuous heat is needed? Hence for continuous on-body sensing this electrode does not seem to be practical. Please add a small discussion addressing this

Our response: For the experiment in this manuscript, we used a heat patch to ensure that the temperature and size of the electrode remained constant, but in practice this patch may not be necessary. We have done experiments simply dropping the TR-CP on human skin (**Fig. R2**) and it remained gelled until washed off with cold water.

Fig. R2: Picture of the TR-CP gelling on contact with skin.

Our changes to the manuscript: We added the following sentence:

“We note that the heating patch may not be necessary in practical applications as the TR-CP gels on contact with warm surfaces, including skin.”

- **Self-adhesiveness: irritation is mentioned as an additional advantage of this electrode vs commercial however irritation comes also from the adhesives used in wearable electrodes and this electrode does not seem to be self-adhesive. Please add a small discussion addressing this**

Our response: We were actually careful not to mention “irritation” in the manuscript as we have not tested it yet. The TR-CP is indeed not adhesive to skin right now. We are working on a new formulation that includes adhesive units. We have added a discussion regarding adhesion in the manuscript.

Our changes to the manuscript: we added the following discussion:

“While conformal, the TR-CP is only weakly adhesive to skin, the dressing was therefore required to maintain the device in place. We are currently working on a new formulation that would be self-adhesive and prevent the use of a potentially irritable tape.”

6. Gel dissolution: Elaborate on the mechanisms of gel dissolution under physiological conditions (e.g., enzymatic activity or ionic strength).

Our response: For now, we can only hypothesize as to the dissolution of the gels. In the in vivo studies, the TR-CP was clearly still present after several days as seen by the blue color. PEDOT:PSS is not biodegradable, so we do not believe it would be sensitive to enzymatic activity. Similarly, we have tested the TR-CP under different pH and ionic conditions and it seems to remain in the gel state under a wide range of conditions. What could happen, however, is a mechanical degradation of the gels where the TR-CP remains a gel (as long as the temperature is maintained) but breaks down into small chunks due to its low toughness. In the manuscript, we now elaborate (see below) on the potential fate of the TR-CP in the alginate composites.

Our changes to the manuscript: The following text has been added to the manuscript.

“We also observed differences in maintenance of the gel volume between the two materials. Alginate is easily degraded in the body due to an ion exchange mechanism that displaces the crosslinking divalent cations.³ This degradation is characterized by a reduction in the volume of the gel after implantation. The addition of TR-CP to the gel prevented this breakdown, as

the volume of the gel was maintained (**Fig. 5e**). Similarly to PEDOT:PSS,⁴ we believe that the TR-CP is non-biodegradable and likely disrupted ion flow thereby preventing the decrosslinking of the alginate.” If the TR-CP was used by itself, we do not believe that it would redissolve as it is stable under a wide range of pH and ionic conditions, but mechanical breakdown is likely due to its low toughness.”

And have added the following references to this section:

(3) Sahoo, D. R.; Biswal, T. Alginate and its application to tissue engineering. *SN Applied Sciences* **2021**, 3 (1), 30. DOI: 10.1007/s42452-020-04096-w.

(4) Seiti, M.; Giuri, A.; Corcione, C. E.; Ferraris, E. Advancements in tailoring PEDOT:PSS properties for bioelectronic applications: A comprehensive review. *Biomaterials Advances* **2023**, 154, 213655. DOI: <https://doi.org/10.1016/j.bioadv.2023.213655>.

7. In vivo cytocompatibility (Section 3.11.1): Specify sample size and statistical methods (no of animals) used in these studies.

Our response: To minimize animal impact, animal sample size was $n = 2$, each animal had $n = 4$ samples implanted for $n = 4$ per condition.

Our changes to the manuscript: We have modified the caption of **Fig. 5**.

The following text was added to Section 3.11.1:

To minimize animal impact, animal sample size was $n = 2$, each animal had $n = 4$ samples implanted for $n = 4$ per condition.

8. Figure S3: Provide clearer interpretation of temperature-dependent rheological behavior, specifically trends in G' and G'' and their relation to molecular dynamics.

Our response: We had previously included a description of the temperature-dependent rheological behavior in the Experimental Design section, just before **Fig. S3**. We have enhanced that section with additional details, such as discussing the trends in G' and G'' and conclusions from these results.

Our changes to the manuscript: The following edits have been made to the text in the SI:

“Experimental design of the PSS-*b*-PNIPAM block copolymer for reversible gelation. TR-CPs with varying molecular weight and ratio of PSS: PNIPAM were synthesized using RAFT polymerization. To observe the thermo-response, we recorded the storage modulus

(G') and loss modulus (G'') as a function of temperature. Trends in G' and G'' can be used to understand the molecular mechanism of thermo-response, and its effect on gelation (or lack thereof). Typically, for liquids, G' is either lower than or close to G'' . As the system reaches closer to a gelation, the G' crosses over G'' , demonstrating the ability of the material to display primarily elastic behavior, which is indicative of gelation. We used this $G' - G''$ crossover to determine the presence of a sol-gel transition. The angular frequency for all these measurements was kept the same (10 rad s^{-1}) to compare the samples under similar shear, and was chosen based on the stability of the dispersions under these conditions. The types of thermo-response (increase in viscosity, formation of a weak gel, etc.) recorded from rheology were in good agreement with our visual observations.

The TR-CP with PSS:PNIPAM 1:0.73 by mass (Table 1, Entry 1) showed an increase in both G' and G'' at $37 \text{ }^\circ\text{C}$, but no crossover. This result indicated that PEDOT:PSS_{95-b}-PNIPAM₁₂₈ displayed a slight increase in modulus, but no gelation at least up to $50 \text{ }^\circ\text{C}$ (**Fig. S3 a-b**). The same trend was observed on recording G' and G'' at $37 \text{ }^\circ\text{C}$. The lack of visible thermo-response was attributed to the low overall molecular weight of the block copolymer, as the thermo-response was likely not strong enough to induce gelation. By doubling the molecular weight of both blocks (Table 1, Entry 2), an increase in modulus was observed at $37 \text{ }^\circ\text{C}$, but gelation only happened at $38 \text{ }^\circ\text{C}$ (**Fig. S3c-d**). Increasing the overall molecular weight indeed induced a stronger thermo-response (as seen by the presence of a crossover point in **Fig. S3c**), and by the higher G' and G'' of PEDOT:PSS_{176-b}-PNIPAM₂₆₈ as compared to PEDOT:PSS_{95-b}-PNIPAM₁₂₈). However, since the gelation occurred above physiological temperatures, we deemed this sample unsuitable for applications in bioelectronics. Thus, to lower the gelation temperature, the molecular weight of the PNIPAM block was increased while maintaining the same PSS block size (**Table 1, Entry 3**). This TR-CP gelled at $36 \text{ }^\circ\text{C}$, however, the gel was not stable at $37 \text{ }^\circ\text{C}$ for prolonged times and at volumes larger than 1 mL (**Fig. S3e-f**). The time sweep at $37 \text{ }^\circ\text{C}$ displayed a steep increase in both G' and G'' , but no crossover was observed, which indicated the formation of a weak gel. The unstable gelation could be attributed to the low percentage of PNIPAM, and subsequently, weak thermo-response in the gel. To address this issue, and to further lower the gelation temperature by strengthening the thermo-response, the PNIPAM block was kept 50 kg mol^{-1} , but we decreased the molecular weight of PSS to 18 kg mol^{-1} (**Table 1, Entry 4**). This TR-CP with a mass ratio of 1:2.3 PSS:PNIPAM showed the formation of a stable, fully reversible gel at $35 \text{ }^\circ\text{C}$, as observed by the $G' - G''$ crossover in the temperature sweep and time sweep.”

9. Electronic characterization:

-Discuss the physical implications of the changes in the circuit elements (e.g., R_c , R_e , and R_i) reported in Tables S1 and S2.

- Why Debye and not Randle circuit modelling? (given the electrically conducting properties of the hydrogel). Please include more discussion on the fitting that was used and the underlying hypothesized mechanisms of conductivity within this hydrogel.

Our response:

- a) Choice of circuit model: The Randle circuit model is mostly used to describe the electrochemical reaction taking place at the electrode-electrolyte interface, typically requiring a three-electrode setup. In contrast, the Debye model is usually used to study the ionic and electronic conduction in the bulk of materials and therefore a two-electrode setup is employed sandwiching the material to test. A more extensive discussion on using the Debye model to study the mixed conduction of materials can be found in this paper by Garbarczyk et. al.⁵ In our work, the goal of applying EIS is to track the evolution of ionic and electronic conduction with the change in the hydrogel composition and physical state. Hence, the two-electrode set-up was picked, and the Debye model was applied. However, if the material was used as an electrode coating material, a three-electrode set-up and the Randle-circuit model would be more appropriate. In the modified Debye circuit model that we have employed, R_e is the resistance from the set-up. C_g is the geometric capacitance of the conductive hydrogel. Q_{dl} is the constant phase element representing the non-ideal double-layer capacitance. These phase elements are incorporated to account for any heterogeneity in the sample or irregularities in capacitance.
- b) Mechanism of conductivity in the TR-CP: From the EIS data, we believe that the gels display both ionic and electronic conductivity. However, they are poorly capacitive. The electronic conductivity is attributed to hole transport in and between PEDOT chains. As we have seen in the cryo-EM, PEDOT chains are present outside the spherical particles and form pi-pi stacked aggregates. As the temperature is increased above the gelation point, PEDOT domains are coming in closer contact which would explain the decrease in R_e upon gelation. Ionic conductivity is attributed to the presence of protons (for the TR-CP dispersed in pure water) consistent with previous reports on PEDOT:PSS conductive hydrogels. When the TR-CP is dispersed in PBS, R_i decreases, simply because the solution is now loaded with additional electrolytes. As the temperature increases past the gel point, the R_i increases, again consistent with lower ion diffusion in gels vs solutions.

Our changes to the manuscript:

The following text has been modified/added to the manuscript:

“To confirm this result, a modified Debye circuit model (**Fig. 2c**), commonly used for PEDOT:PSS-based hydrogels²⁴⁻²⁶ was used to extract the electronic resistance (R_e) and ionic resistance arising from the movement of H^+ ions in the absence of added electrolyte (R_i) (**Fig. S7c**). We chose the Debye circuit model as it is best suited for studying mixed conduction in materials sandwiched between two electrodes.^{27, 28} We observed that on increasing temperature, the R_i increased from 86.8 Ω for the liquid to 121.7 Ω for the gel (**Table S1**). We believe that this increase occurs due to the hydrophobic nature of PNIPAM, and the formation of physical crosslinks that hinder ion diffusion in the matrix. Further, R_e decreased from 41.1 k Ω (liquid) to 22.5 k Ω (gel) on increasing temperature (**Table S1**), which is presumably responsible for the significant impedance drop observed in the low-frequency regime (**Fig.S7b**). A two-point probe measurement confirmed a similar decreasing trend in resistance, from 38.1 k Ω to 20.1 k Ω upon gelation, indicating that the Debye model accurately reflects the physical and chemical properties of the TR-CP gel. The observed reduction in electronic resistance after gelation suggests the formation of a more electronically conductive pathway in the gel, likely resulting from closer packing of PEDOT chains. Despite this reduction, the R_e of the formed gel was nearly two orders of magnitude higher than previously reported conductive hydrogels characterized with the same setup and model ($\sim 200 \Omega$).^{17, 25} In order to attain comparable R_e , we investigated the effect of the PEDOT loading and TR-CP concentration on R_e , as we expect these parameters to increase the potential for PEDOT chains overlap and therefore achieve higher electronic conductivity.”

The following text has been added to the Supporting Information, in the Electronic Characterization section:

“The Debye model was used to study the ionic and electronic conduction in the bulk of materials, using the two-electrode system, which is employed through sandwiching the hydrogel between Cu electrodes. In our work, the goal of applying EIS is to track the evolution of ionic and electronic conduction along with the change in the hydrogel composition and physical state. Hence, the two-electrode set-up was picked and the Debye model was applied. In the modified Debye circuit model that we have employed, R_c is the resistance from the set-up, and thus, remains similar for the liquid and gel. C_g is the geometric capacitance of the conductive hydrogel. Q_{dl} is the constant phase element representing a non-ideal double-layer capacitance. These phase elements are incorporated to account for any heterogeneity in the sample or irregularities in capacitance.”

The following references were added to support this section:

Feig, V. R.; Tran, H.; Lee, M.; Bao, Z. Mechanically tunable conductive interpenetrating network hydrogels that mimic the elastic moduli of biological tissue. *Nat. Commun.* **2018**, *9* (1), 2740-2740. DOI: 10.1038/s41467-018-05222-4.

Nguyen, D. M.; Lo, C. Y.; Guo, T.; Choi, T.; Sundar, S.; Swain, Z.; Wu, Y.; Dhong, C.; Kayser, L. V. One Pot Photomediated Formation of Electrically Conductive Hydrogels. *ACS Polymers Au* **2024**, *4* (1), 34-44. DOI: 10.1021/ACSPOLYMERSAU.3C00031.

Lu, B.; Yuk, H.; Lin, S.; Jian, N.; Qu, K.; Xu, J.; Zhao, X. Pure PEDOT:PSS hydrogels. *Nat. Commun.* **2019**, *10* (1), 1043-1043. DOI: 10.1038/s41467-019-09003-5.

Huggins, R. A. Simple method to determine electronic and ionic components of the conductivity in mixed conductors a review. *Ionics* **2002**, *8* (3), 300-313. DOI: 10.1007/BF02376083.

Garbarczyk, J. E.; Wasiucioneck, M.; Machowski, P.; Jakubowski, W. Transition from ionic to electronic conduction in silver–vanadate–phosphate glasses. *Solid State Ionics* **1999**, *119* (1), 9-14. DOI: [https://doi.org/10.1016/S0167-2738\(98\)00475-5](https://doi.org/10.1016/S0167-2738(98)00475-5).

10. Cryo-EM methodology (Section 1.7): Address potential artifacts arising from sample preparation.

Our response: Artefacts are indeed a potential concern, but we took all possible steps to ensure they are minimized and do not believe they interfere with our interpretation. We have now included more details in the Supporting Information (Methods) on reporting defocus value and how we prevented common artefacts in cryo-EM imaging. Specifically, all samples were prepared under 100% RH to prevent drying, and we prepared multiple samples for each condition to enable the exclusion of samples with visible water ice formation.

Our changes to the manuscript: The following text has been added to the manuscript:

“We confirm the lamellae visible in cryo-EM images are not due to artefacts from the oscillating contrast transfer function of the microscope by demonstrating that the defocus value does not affect the lamellae spacing (**Fig S13**).”

The following text has been added to the Supporting Information in the Methods section:

“We minimized the potential for artefacts from water evaporation by preparing samples under 100% relative humidity, to maintain sample hydration during blotting and prior to plunge freezing. Although ice formation can sometimes occur, we prepared 3–5 grids per temperature condition, and grids showing evidence of crystalline ice were excluded from imaging.”

“The gelation of the TR-CP at higher temperatures led to thick layers that are challenging to image. For 45 °C samples, the Vitrobot chamber was heated to 45 °C while the solution was kept at 37 °C. We used solutions at 37 °C, as they allow the formation of vitreous ice layers that are thin enough for TEM imaging (< 100 nm) while maintaining the gelation behavior of the TR-CP.”

The following figure was added to the SI:

Fig. S13. Cryo-EM micrographs of 1.8 wt% TR-CP at three different defocus values. Cryo-EM micrograph with large, **a**, medium, **b**, and small **c**, defocus values. Panels **d**, **e**, and **f**, show the zoomed-in view of blue boxes shown in **a**, **b**, and **c**, respectively. The white arrows point to lamella fringes observable in these micrographs. **g**, shows radial integration of FFT of micrographs shown in panel **a**, **b**, and **c** demonstrating the change in defocus and perturbations to the contrast transfer function (CTF). **h**, line scans of the lattice fringes. The peak-to-peak distances are consistent with the lamella spacing of PEDOT:PSS and does not change with defocus value. These images highlight that the lattice fringes observed in cryo-EM micrographs of TR-CP samples are from PEDOT:PSS chain stacking, and are not an artefact from the oscillation in the CTF (compare **g**, **h**). The azimuthally integrated FFT profiles (**g**) show no distinguishable peak due to PEDOT:PSS chain stacking (6 nm^{-1} , 1.64 nm) due to the low density of features and low number of stacked chains per feature.

11. Units and symbols: Standardize units across sections (e.g., mS/cm vs. $\text{mS}\cdot\text{cm}^{-1}$).

Our response: We have edited the manuscript to standardize the unit to mS cm^{-1} throughout.

12. **In vivo biocompatibility: Is there any in vivo degradation analysis over time? -**

Our response: For the scope of this paper, we limited the in vivo biocompatibility study to 2 weeks. We observed that while alginate degraded in the body, the TR-CP/alginate maintained its structural integrity. However, the in vivo degradation of the TR-CP is certainly of interest and will need to be studied extensively before the materials can be adopted for in vivo bioelectronic interfacing.

Our changes to the manuscript: We added the following text:

We also observed differences in maintenance of the gel volume between the two materials. Alginate is easily degraded in the body due to an ion exchange mechanism that displaces the crosslinking divalent cations.³⁶ This degradation is characterized by a reduction in the volume of the gel after implantation. The addition of TR-CP to the gel prevented this breakdown, as the volume of the gel was maintained (**Fig. 5e**). Similarly to PEDOT:PSS,¹⁹ we believe that the TR-CP is non-biodegradable and likely disrupted ion flow thereby preventing the decrosslinking of the alginate.

13. **Control experiments: Is there any control with gels formed with PNIPAM alone and covalently crosslinked PEDOT:PSS ? to validate the superiority of TR-CPs.**

Our response: We initially wanted to perform such a control, however, we could not find a feasible material system. We could not find a system where we could get a sol-gel transition without using our TR-CP. Any system incorporating PNIPAM had to be permanently crosslinked, in which case it did not have a sol-gel transition. When no covalent linking between PNIPAM and PEDOT or PSS was present, no gelation was obtained due to phase separation. In itself, the fact that the block copolymer structure is needed to get the sol-gel behavior is a superior attribute. For the in vivo and in vitro studies, we are not claiming that our material is superior but rather that it is cytocompatible, and could be used in biological applications.

14. **Figure captions: Enhance captions with details. For instance:**

- Fig. 2: Explain how PEDOT loading affects ionic conductivity.
- Fig. 5: Describe observed microvascularization in histological images.
- Table 1: Justify why the specific mass ratio of PSS:PNIPAM selected for gelation properties.

Our response: The following additions have been made. We have not added the explanation for the ionic conductivity in Fig 2 because the discussion is too long for the caption. We did not edit the caption for Table 1, as we have included a column showing that only one TR-CP (Entry 4) forms a stable gel.

Our changes to the manuscript: The following edits/additions have been made:

Caption for Fig. 2a: “Bode plot obtained by EIS showing the effect of PEDOT loading on the impedance of the gels (3.6 wt% in DI water). The TR-CP with SS:NIPAM:EDOT ratio of 3.5:16:1 showed the lowest impedance.”

Caption for Fig. 5d – i: “Hematoxylin-eosin staining of skin sections with implanted gels at 1 and 2 weeks post implantation where G indicates gel area visible as dark purple color and arrow points to microvasculature observed in the TR-CP/alginate samples, which appear as small ovoid vessels containing red-stained erythrocytes.”

15. Repeated exposure: Discuss whether repeated thermal cycling of gels could lead to cellular or tissue damage, particularly due to material degradation or residual chemicals.

Our response: We have not observed any changes to the mechanical properties of the TR-CP after repeated heat-cool cycles, except some dehydration which are observed after $n = 4$ cycles on the rheometer. Both PNIPAM and PEDOT:PSS are widely used in biomedical applications due to their ability to withstand physiological temperature without degradation. Additionally, to the best of our knowledge, there are no residual small molecules in the TR-CP due to our fully polymeric system and the purification procedure, which involves the removal of salts with acidic and basic resins, and dialysis. Regarding material degradation, as we show the materials are self-healing close to the LCST so if thermal cycling occurs these materials should be repaired and not damage tissue or cells. Any further discussion on thermal cycling would be speculative as we have not done these tests.

Our changes to the manuscript: We added the following sentence:

“For the scope of this study, we have not investigated the effects of repeated thermal cycling on cells and tissues, but mechanical and electronic studies have shown that the properties of the materials are stable upon cycling.”

16. Typos: Line 271 states, “he TR-CPs were then incorporated...” This appears to be a typo and should be corrected to “the TR-CPs.”

Our response: We have fixed this typo in the text.

These revisions, are sought to strengthen the manuscript's impact and make it further suitable for publication.

Reviewer #2 (Remarks to the Author):

Damani et al. did a systematic study of the polymer hybrid, covering from design through synthesis and characterization to their applications in bioelectronics. The story is very interesting and should interest broad readers in the fields of chemistry, materials science and bioelectronics.

Our response: Thank you for reviewing our manuscript.

The following points should be addressed in the revision, to improve the clarity of the story.

1. One major concern is the structure information of the PEDOT:PSS-PNIPAM, based on SAXS and Cryo-EM. In general, it seems that they are not consistent with each other. Those nanostructures like lamellae or particle, as mentioned in page 8 (lamellar fringes with ~1.6 nm spacing and small particles (~5 nm)) are not observed in the SAXS data. Although the SAXS data fitting based on Guinier-Porod model could provide some structure information. Technically, the fitting procedure is blurred by the confusing descriptions: on one side, the authors mentioned the reference measurement and background correction in the SAXS experiment section; on the other side, “Two peaks (Figure S13) in the high q region ($q = 15 \text{ nm}^{-1}$ and $q = 28 \text{ nm}^{-1}$) were attributed to the sample holder cells and water background, and excluded from the fitting.”, which shows clearly these two peaks were not subtracted for the fitting (Figure S13 of SI). Meanwhile, how were the two Guinier-Porod models were used for the data fitting? Is this method reliable for such a complex system? Correlations with the cryo-EM data plays vital role.

Our response: We agree that the microstructure discussion lacked clarity. We hope the changes below improve the manuscript. The discrepancy between SAXS and cryo-EM is because the lamellar peaks of PEDOT stacking are not visible in the SAXS. Cryo-EM reveals that PEDOT:PSS nanocrystals consist of only a few lamellar stacks (2–4 chains), which would result in a broad scattering peak difficult

to detect using SAXS. The absence of a SAXS peak that corresponds to the lamellar spacing (~ 1.6 nm) is expected given the low number of these domains and the small domain size (which leads to peak broadening). Indeed, these features are not apparent when FFTs of the cryo-EM images are calculated despite being clearly visible. We confirm the structural origin of these features by taking images at different defocus values, showing that imaging artefacts are not responsible for the appearance of these domains (**Fig. S13**, added below).

The peak centered at 5.6 nm^{-1} is attributed to the block copolymer (See **Fig. R3**, to be included in a future manuscript). For the SAXS fitting, we now used a core-shell sphere model to describe the spherical nanoparticles, with a PEDOT core and a PSS-PNIPAM shell. From the new model, the core-shell spheres have total radius (core radius plus the shell thickness) of ~ 4 nm and 5 nm below and above the LCST, respectively. Details of the fitting have been added to the main text and SI. The peaks in the high q region are from WAXS detector and are now excluded from the plot.

Fig. R3: Change in intensity as a function of q . Intensity profiles were recorded at different temperatures to capture the change in shape due to the LCST. Sample concentration = 10 mg/mL in DI water.

Further, we have re-plotted the figures related to the core-shell sphere model fitting in the SI to exclude data beyond 10 nm^{-1} to avoid confusion with the peaks arising from the sample cell holder and background.

Our changes to the manuscript: We have changed the description of SAXS fitting to the following: “To confirm our observations from microscopy, small-angle X-ray scattering (SAXS) at 20 °C and 40 °C in solution was used to study the colloidal particle self-assembly, gel structure, and gelation mechanism (**Fig. 3e**, additional details in **SI Section 3.5**). A core-shell sphere model fitting²⁹ revealed that the TR-CP colloids in solution

below the LCST consist of 3.2 nm PEDOT cores surrounded by a 0.9 nm PSS-PNIPAM shell, very close to the size of the spherical/ovoid nanoparticles seen by cryo-EM (**Fig. S16, Table S4**). The peak centered at 5.6 nm^{-1} is attributed to the intermolecular interactions from the PNIPAM. Two peaks in the high q region ($q = 15 \text{ nm}^{-1}$ and $q = 28 \text{ nm}^{-1}$) are attributed to the sample holder cells and water background, and excluded from the fitting. As the temperature was increased above gelation, both the core radius and the shell thickness of these spherical nanoparticles slightly increased to 3.9 nm and 1.3 nm, respectively. The interparticle distance (effective radius) of the nanospheres decreased from 12.2 nm to 5.8 nm above the LCST, consistent with the transition into a gel state through particle aggregation. We note that the SAXS did not capture the presence of the PEDOT lamella seen by cryo-EM due to their low concentration and overlap with background scattering.”

We have also revised the fitting method in Supporting Information, Section 3.5.

We replaced the figure in the SI with the following figure, excluding data beyond 10 nm^{-1} to avoid confusion with the peaks arising from the sample cell holder and background.

Fig. S16. a. Core shell sphere model on Small-angle x-ray scattering results on TR-CP showing measurement, fitting lines and total fit **b.** below LCST and **c.** above LCST.

- It is recommended that the authors should include more evidences to support their judgements like “These lamellar fringes are likely associated with a semi-crystalline PEDOT phase. The small size of these particles in solution likely explains the excellent colloidal stability of the dispersion and their ability to be lyophilized and redispersed.”. If the PEDOT was crystallized, which should also be observed with the SAXS data? Did the authors cross-check and exclude the artefacts caused by ice formation, for the cryo-EM data?

Our response: To confirm that the lattice fringes in cryo-EM have a structural origin and are not due to an imaging artefact, we acquired TEM images at different defocus values (Fig. S13). We observe that although we vary the CTF by changing the defocus, the spacing of the lamellae fringes does not change, confirming that

these fringes are due to chain stacking. As PSS and PNIPAM are generally amorphous polymers, these well-ordered features can only be attributed to PEDOT. They are too large to be pi-pi stacking but are consistent with previous distances reported for PEDOT lamella stacking. Again, we do not expect that SAXS would show a peak associated with these features, given their low number density and small domain size. Indeed, these features are not apparent when FFTs of the cryo-EM images are calculated despite being clearly visible.

A discussion on ice artifacts was also added in response to reviewer 1 (see above). As will be reported in a separate article, we find the presence of these lamellar structures to be also present in PEDOT:PSS, especially once salt additives are present, as we show below:

REDACTED

Our changes to the manuscript: We added Fig. S13 and modified the SI (Cryo-EM sample preparation section) to address the reviewer's concerns.

Fig. S13 Cryo-EM micrographs of 1.8 wt% TR-CP at three different defocus values. Cryo-EM micrograph with large, **a**, medium, **b**, and small **c**, defocus values. Panels **d**, **e**, and **f**, show the zoomed-in view of blue boxes shown in **a**, **b**, and **c**, respectively. The white arrows point to lamella fringes observable in these micrographs. **g**, shows radial integration of FFT of micrographs shown in panel **a**, **b**, and **c** demonstrating

the change in defocus and perturbations to the contrast transfer function (CTF). **h**, line scans of the lattice fringes. The peak-to-peak distances are consistent with the lamella spacing of PEDOT:PSS and does not change with defocus value. These images highlight that the lattice fringes observed in cryo-EM micrographs of TR-CP samples are from PEDOT:PSS chain stacking, and are not an artefact from the oscillation in the CTF (compare **g**, **h**). The azimuthally integrated FFT profiles (**g**) show no distinguishable peak due to PEDOT:PSS chain stacking (6 nm^{-1} , 1.64 nm) due to the low density of features and low number of stacked chains per feature. This is consistent with our SAXS results, which do not capture the presence of PEDOT:PSS lamella spacing as expected.

3. It's a bit confusing about the description of the colloidal behavior of the PEDOT:PSS-PNIPAM. If the authors believe it was a colloid, then please provide the structure information such as size and shape. In addition, it is important to illustrate the dominating factors for their stable dispersion in the solvent. It should be helpful to understand the structure-property correlations if the authors could clarify the colloidal and hydrogel behavior of the PEDOT:PSS-PNIPAM.

Our response: As explained in the previous comment, we used a core-shell model to fit the SAXS curve and correlate with observations from cryo-EM. According to the fitting, the particles are spherical and consist of a PEDOT core with a radius of 3.2 nm, surrounded by a 0.9 nm shell, most likely PSS-PNIPAM, below the LCST. Above the LCST, the PEDOT core radius increases to 3.9 nm, with a surrounding 1.3 nm PSS-PNIPAM shell. We refer to these spherical structures as colloids because that is the most commonly used term to describe PEDOT:PSS particles in solution.

Our changes to the manuscript: We have changed the description of SAXS fitting. See text above in comment 1.

4. It is not very clear why the authors include a series of PSS-b-PNIPAM samples. The logic connection with the main story should be clarified in the main text.

Our response: We synthesized several PSS-*b*-PNIPAM samples with varying molecular weights and mass ratio of PSS:PNIPAM to determine the best molecular composition for a reversible sol-gel transition under 37 °C, as seen in Table 1. We have clarified in the main text and included more details in the Experimental Design section of the Supporting Information (see response to Reviewer 1).

Our changes to the manuscript: The following text has been added to the manuscript:

“We synthesized a series of PEDOT:PSS-*b*-PNIPAM with varying molecular weights and block ratio (**Table 1**) to study the effect of molecular weight on the thermo-response, and to

determine the optimal ratio of PSS:PNIPAM to enable the formation of stable gels at physiological temperature.”

5. Did the authors remove the residue of the catalysis and oxidant during the synthesis of the PEDOT:PSS-PNIPAM. In principle, $\text{FeCl}_3/\text{Na}_2\text{S}_2\text{O}_8$ and their derivatives would influence the electronic and ionic properties of the sample. It could also bring challenges for the conductivity measurements since, the copper used in electrical conductivity (e.g., Figure S6) measurement might react with FeCl_3 , for example. In addition, these residues might bring negative influence for their bioelectronic applications. Please clarify these points.

Our response: As mentioned on Page S10 of the Supplementary Information, “PEDOT:PSS-*b*-PNIPAM was purified by simultaneously stirring over 3mL of acidic resin (Dowex Marathon C) and 2mL basic (Lewatit Ion Exchange) resin.” The acidic resin is responsible for the removal of excess iron by cation exchange, and the basic resin removes the $\text{Na}_2\text{S}_2\text{O}_8$ reduction by-products. We collected x-ray photoelectron spectroscopy (XPS) spectrum (**Fig. R5**) showing that there are no residual peaks arising from Fe in the purified samples (706.7 eV to 710.7 eV).

Fig. R5: X-ray photoelectron spectroscopy shows no peaks in the 706.7 eV to 710.7 eV range, which would correspond with the presence of Fe.

6. It should be helpful if the authors could illustrate the dominating factors for the electronic and ionic conductivities of the PEDOT:PSS-PNIPAM. Based on which, discussions about the changes of the conductivities as a result of temperature change, in the section of “Electronic properties of the TR-CP”. In addition, correlations with the microstructures mentioned in the following section should help the readers to understand the story better.

Our response: Reviewer 1 asked for similar clarification. The new discussion on this topic is copy-pasted below.

Our changes to the manuscript: The following edits have been made to the manuscript:

“To confirm this result, a modified Debye circuit model (**Fig. 2c**), commonly used for PEDOT:PSS-based hydrogels²⁴⁻²⁶ was used to extract the electronic resistance (R_e) and ionic resistance arising from the movement of H^+ ions in the absence of added electrolyte (R_i) (**Fig. S7c**). We chose the Debye circuit model as it is best suited for studying mixed conduction in materials sandwiched between two electrodes.^{27, 28} We observed that on increasing temperature, the R_i increased from 86.8 Ω for the liquid to 121.7 Ω for the gel (**Table S1**). We believe that this increase occurs due to the hydrophobic nature of PNIPAM, and the formation of physical crosslinks that hinder ion diffusion in the matrix. Further, R_e decreased from 41.1 k Ω (liquid) to 22.5 k Ω (gel) on increasing temperature (**Table S1**), which is presumably responsible for the significant impedance drop observed in the low-frequency regime (**Fig.S7b**). A two-point probe measurement confirmed a similar decreasing trend in resistance, from 38.1 k Ω to 20.1 k Ω upon gelation, indicating that the Debye model accurately reflects the physical and chemical properties of the TR-CP gel. The observed reduction in electronic resistance after gelation suggests the formation of a more electronically conductive pathway in the gel, likely resulting from closer packing of PEDOT chains. Despite this reduction, the R_e of the formed gel was nearly two orders of magnitude higher than previously reported conductive hydrogels characterized with the same setup and model ($\sim 200 \Omega$).^{17, 25} In order to attain comparable R_e , we investigated the effect of the PEDOT loading and TR-CP concentration on R_e , as we expect these parameters to increase the potential for PEDOT chains overlap and therefore achieve higher electronic conductivity.”

7. It would be very helpful if the authors could include more related references, especially in the results and discussion part.

Our response: We have now included a few more relevant citations in the manuscript, particularly in the Electronics, Microstructure, and Cytocompatibility discussions.

8. Line 186, P7, the authors wrote that “To elucidate the nanoscale structure of the TR-CP and what prompts it to form a porous network,”. What is the porous network mean?

Our changes to the manuscript:

We have changed the sentence to “To elucidate the nanoscale structure of the TR-CP and what prompts it to form a gel,...”.

9. It should be a mistake to use the terms like molar ratio of PSS:PEDOT molar ratio. In principle, the authors could get the numbers of SS units and the EDOT that used for the synthesis. Meanwhile, do the authors have the Mw information of PEDOT?

Our response: We agree and have made changes throughout to refer to SS:EDOT ratio.

Unfortunately, it was not possible to obtain the molecular weight of PEDOT as it is too insoluble to be ran on GPC and does not fly well on MALDI-TOF.

Our changes to the manuscript: SS:EDOT changed throughout.

10. Please clarify the sentence in the SI, “We chose 37 °C for solution temperature as it allows the formation of vitreous ice layers which are thin enough for TEM imaging while maintaining the gelation behavior of the block copolymer.”.

Our response: We understand the confusion and have updated the SI to clarify our sample preparation.

Our changes to the manuscript: We have modified the SI to:

“The gelation of the TR-CP at higher temperatures leads to thick layers that are challenging to image. For 45 °C samples, the Vitrobot chamber was heated to 45 °C while the solution was kept at 37 °C. We used solutions at 37 °C, as they allow the formation of vitreous ice layers that are thin enough for TEM imaging (< 100 nm) while maintaining the gelation behavior of the block copolymer. The Quantifoil grids were then equilibrated at 45 °C for at least 10 minutes.”

Reviewer #3 (Remarks to the Author):

This manuscript report a new conductive thermos responsive hydrogel with reversible gelation based on a PEDOT:PSS poly(N-isopropylacrylamide) (PSS-b-PNIPAM) complex.

The reversible gelation occurs at 35 degrees celsius. Overall, the manuscript gives a very detailed description of the synthesis and the characterization of the hydrogel. However, its relevance for the material science community is not very well substantiated. As I do not see any potential breakthrough in terms of materials, I suggest rejection and publication in a more specialized journal.

Our response: Thank you for reviewing our manuscript. We believe that our manuscript provides inspiration for the chemical modification of PEDOT:PSS to achieve unique properties, in this case a sol-gel transition which is otherwise not possible. We believe that given the popularity of PEDOT:PSS and the need for new stimuli-responsive conductive polymers in bioelectronics, it will be of broad appeal to several different communities from materials science to electronics.

Please find attitional comments below.

1. By which mechanism does the impedance reduction of TR-CP occur due to temperature changes and consequently upon gelation? According to the numbers given in page 5 and table S1, the effect on the temperature on electronic and ionic resistance is comparable, with electronic resistance decreasing by about 50% and ionic resistance increasing by about 30%. Is unclear why the effect on ionic conductivity is not discussed. Also, in the tables S1-S3 the values are given with several significant digits without error bars.

Our response: Reviewers 1 and 2 have raised a similar question. We have copy-pasted below our changes to the manuscript addressing it. We have also reduced the number of significant digits.

Our changes to the manuscript: The following text has been added to the manuscript:

“To confirm this result, a modified Debye circuit model (**Fig. 2c**), commonly used for PEDOT:PSS-based hydrogels²⁴⁻²⁶ was used to extract the electronic resistance (R_e) and ionic resistance arising from the movement of H^+ ions in the absence of added electrolyte (R_i) (**Fig. S7c**). We chose the Debye circuit model as it is best suited for studying mixed conduction in materials sandwiched between two electrodes.^{27, 28} We observed that on increasing temperature, the R_i increased from 86.8 Ω for the liquid to 121.7 Ω for the gel (**Table S1**). We believe that this increase occurs due to the hydrophobic nature of PNIPAM, and the formation of physical crosslinks that hinder ion diffusion in the matrix. Further, R_e decreased from 41.1 k Ω (liquid) to 22.5 k Ω (gel) on increasing temperature (**Table S1**), which is presumably responsible for the significant impedance drop observed in the low-frequency regime (**Fig.S7b**). A two-point probe measurement confirmed a similar

decreasing trend in resistance, from 38.1 k Ω to 20.1 k Ω upon gelation, indicating that the Debye model accurately reflects the physical and chemical properties of the TR-CP gel. The observed reduction in electronic resistance after gelation suggests the formation of a more electronically conductive pathway in the gel, likely resulting from closer packing of PEDOT chains. Despite this reduction, the R_e of the formed gel was nearly two orders of magnitude higher than previously reported conductive hydrogels characterized with the same setup and model ($\sim 200 \Omega$).^{17, 25} In order to attain comparable R_e , we investigated the effect of the PEDOT loading and TR-CP concentration on R_e , as we expect these parameters to increase the potential for PEDOT chains overlap and therefore achieve higher electronic conductivity.”

2. The lowest impedance was observed when the PSS:PEDOT ratio was 3.5:1, how can the conduction mechanism of PEDOT:PSS be explained, considering the difference between ionic conductivity and electronic conductivity?

Our response: We have included additional details about the mechanism of conductivity and changes in ionic and electronic conductivity in the changes above.

Our changes to the manuscript: In addition to the changes detailed above, we have made the following edit to the text:

“We found that the 3.5:1 ratio showed the lowest impedance (**Fig. 2a**), which we presume is due to the most homogenous dispersion of PEDOT chains.”

3. The ionic conductivity refers to which ion?

Our response: The polyelectrolyte complex contains H⁺ ions, originating from the sulfonate group present on the PSSH. Some residual Na⁺ ions may be present, however it is unlikely given the pK_a of PSS and the pH recorded at 2. For the samples dispersed in PBS, which is a blend of NaCl, KCl, KH₂PO₄ and Na₂HPO₄, we expect mobile ions (K⁺, PO₄³⁻, Cl⁻) to participate in the ionic conductivity.

Our changes to the manuscript: We added the following text to the manuscript:

“To confirm this result, a modified Debye circuit model (**Fig. 2c**), commonly used for PEDOT:PSS-based hydrogels²⁴⁻²⁶ was used to extract the electronic resistance (R_e) and ionic resistance arising from the movement of H⁺ ions in the absence of added electrolyte (R_i) (**Fig. S7c**).”

4. Figure 2d: is the circuit closed inly upon gelation?

Our response: The LED is on faintly when the TR-CP is in the liquid state, but the light is stronger when gelled.

5. 2f: the caption says: “Nyquist plot for the TR-CP at 5.6 wt% in PBS (pH = 4) which showed the highest conductivity.” It is unclear what shows the highest conductivity.

Our response: Fig. 2f shows the Nyquist plot for the 5.6 wt% TR-CP in PBS at pH = 4, which is the TR-CP with the highest conductivity out of all the formulations that we tested.

Our changes to the manuscript: We have edited the caption for clarity:

“f. Nyquist plot for the TR-CP at 5.6 wt% in PBS (pH = 4). This formulation shows the highest conductivity among all the TR-CP samples.”

6. Why did the conductivity decrease when the concentration increased to 7.6 wt%? Can you explain in more detail?

Our response: Visually, we observed that the 7.6 wt% TR-CP did not disperse well as compared to the 3.6 wt% and 5.6 wt% TR-CP, and showed the presence of clumps or aggregates. We believe that these solid clumps hinder the diffusion of ions. Additionally, the gel is not continuous anymore which limits electronic conduction. The non-homogeneity of the sample would also cause disruptions to contact with the electrode. The intermittent nature of conductive pathways in the 7.6 wt% sample is further confirmed by the presence of irregular signals in the Nyquist plot for this sample (**Fig. S8b**)

Our changes to the manuscript: We have added the following text to the Supporting Information:

“Decrease in conductivity on increasing concentration above 5.6 wt%: Visually, we observed that the 7.6 wt% TR-CP did not disperse well as compared to the 3.6 wt% and 5.6 wt% TR-CP, and showed the presence of clumps or aggregates. We believe that these solid clumps hinder the diffusion of ions. Additionally, the gel is not continuous anymore which limits electronic conduction. The non-homogeneity of the sample would also cause disruptions to contact with the electrode.”

7. The conductivity calculation results of 5.6wt% TR-CP ($\sigma_i = 200 \text{ mS cm}^{-1}$, $\sigma_e = 14 \text{ mS cm}^{-1}$) was mentioned to be similar to those of existing conductive hydrogels. However, did this study specify which specific conductive hydrogel was compared?

Our response: This statement was in comparison with PEDOT:PSS-based hydrogels which have been developed for use as bioelectronic interfaces (i.e., without potentially toxic secondary dopants). We have now specified in the manuscript.

Our changes to the manuscript: The following change has been made to the text:
“The highest conductivity recorded was for 5.6 wt% TR-CP (PSS:PNIPAM:PEDOT ratio of 3.5:16:1) in PBS: $\sigma_i = 200 \text{ mS cm}^{-1}$ and $\sigma_e = 14 \text{ mS cm}^{-1}$ from EIS (**Fig. 2f**) and $\sigma = 10.4 \text{ mS cm}^{-1}$ from 2-point probe, which is on par with other previously reported conductive hydrogels based on PEDOT:PSS and formulated for bioelectronic interfaces.”

7. Why did the authors choose a two-point probe method instead of a four-point probe for measuring conductivity in the TR-CP samples?

Our response: The four point probe method is not reliable for such soft hydrogels, because the thickness of the sample changes as the probe presses down which would change the conductivity calculation. On the other hand, the method we have previously developed using 2 electrodes in a cuvette allows us to maintain precisely the geometry for most accurate measurements.

Our changes to the manuscript: The following text was added to the Supporting Information:

“Here, we have used a two-point probe for our measurements. The four point probe method is not reliable for such soft hydrogels, because the thickness of the sample changes as the probe presses down which would change the conductivity calculation. On the other hand, the method we have previously developed using 2 electrodes in a cuvette allows us to maintain precisely the geometry for most accurate measurements. For the electronic measurements on samples with variable PEDOT concentration, 1 cm^3 of TR-CP was used. For all other electronic measurements, sample volumes and solvent conditions are as specified in captions.

The Debye model was used to study the ionic and electronic conduction in the bulk of materials, using the two-electrode system, which is employed through sandwiching the

hydrogel between Cu electrodes. In our work, the goal of applying EIS is to track the evolution of ionic and electronic conduction along with the change in the hydrogel composition and physical state. Hence, the two-electrode set-up was picked and the Debye model was applied. In the modified Debye circuit model that we have employed, R_c is the resistance from the set-up, and thus, remains similar for the liquid and gel. C_g is the geometric capacitance of the conductive hydrogel. Q_{dl} is the constant phase element representing a non-ideal double-layer capacitance. These phase elements are incorporated to account for any heterogeneity in the sample or irregularities in capacitance.”

And we have added the following reference:

Daso, R. E.; Posey, R.; Garza, H.; Perry, A.; Petersen, C.; Fritz, A. C.; Rivnay, J.; Tropp, J. Standardized Electrochemical Characterization of Conductive Hydrogels. 2025. DOI: 10.26434/chemrxiv-2025-6n1bn.

8. The concentration of 3.6 wt% was determined to be the lower concentration for gel formation. Could authors explain what changes occurred in the microstructure of TR-CP as the concentration increased?

Our response: To understand the differences in the gelation mechanism for the 3.6 wt% and 5.6 wt% samples, we analyzed the plots for the storage modulus as a function of time at 37 °C. We have detailed our observations below.

Our changes to the manuscript: The following text and figure have been added to the Supporting Information:

“A close look at the storage modulus of the 5.6 wt% and 3.6 wt% TR-CP during the time sweep (**Fig. S15**) revealed subtle differences in the heating profiles. While the 3.6 wt% TR-CP showed just one slope (= 0.55), the 5.6 wt% displayed two slopes – one, similar to the 3.6 wt% TR-CP at 0.81, and a second slower heating regime with a much higher slope of 5.18. This difference may suggest the occurrence of a ‘pre-gel’ state in the 5.6 wt% before the formation of a fully interconnected network.”

Fig. S15. Change in storage modulus as a function of time at 37 °C, for 3.6 and 5.6 wt% TR-CP. Analysis of the slopes show that the 5.6 wt% TR-CP has two crosslinking ‘events’ that lead to the formation of a stable gel, which may hint at differences in the mechanism of crosslinking for different concentrations.

9. The statement “...an increased inflammatory response is common when conducting polymers are implanted” should be used with caution. This is supported only by two quite old citations, while many other studies do not show similar evidence.

Our response: We have toned down this statement and added more recent references.

Our changes to the manuscript: The sentence was changed to:

“As others have observed, an increased inflammatory response is possible when conducting polymers are implanted.”

New references:

Liu, Y.; Feig, V. R.; Bao, Z. Conjugated Polymer for Implantable Electronics toward Clinical Application. *Advanced Healthcare Materials* **2021**, *10* (17), 2001916. DOI: <https://doi.org/10.1002/adhm.202001916>.

Li, G.; Huang, K.; Deng, J.; Guo, M.; Cai, M.; Zhang, Y.; Guo, C. F. Highly Conducting and Stretchable Double-Network Hydrogel for Soft Bioelectronics. *Advanced Materials* **2022**, *34* (15), 2200261. DOI: <https://doi.org/10.1002/adma.202200261>.

10. For the application point of view, the advantage of the material remains unclear both for invasive and non-invasive bioelectronics. The EMG epidermal electrodes do not seem to have a particular advantage with respect to similar materials based on PEDOT which are also easier to fabricate.

Our response: The focus of this manuscript was on fundamental materials science describing a new material with unique sol-gel and conducting properties. We agree that the EMG may be more difficult to fabricate than other systems, and therefore may not illustrate best the power of the TR-CP. But, if the TR-CP was gelled around hairy skin or scalp or directly injected (which we did not show herein), it would have clear advantages in terms of conformability and non-invasiveness. Further demonstrations of applications are ongoing and will be the topic of future manuscripts.

11. The relevance of self-healing properties is unclear. Also, it is unclear the time required for the process and how mechanical and electrical properties change before and after healing.

Our response: As we have shown in the sections preceding the self-healing, the mechanical and electronic properties are preserved over many heat-cool cycles (analogous to the self-healing process). In the case of the experiment shown in Fig. 6a, the healing process took 25 – 30s to heal for a volume of 0.6 – 1mL.

Our changes to the manuscript: The following text has been added to the caption of **Fig. 6a**:

“The gels took ~25 s to heal for a volume of 0.6 mL.”

12. In terms of cytocompatibility, the systems behaves similarly to other hydrogels and electrode materials based on PEDOT:PSS. Also explain why that type of cell was selected.

Our response: Yes, the cytocompatibility of the TR-CP is similar to other hydrogels based on PEDOT:PSS. As mentioned in the manuscript and SI, the International Organization for Standardization (ISO) standard 10993-5 is a standard used to evaluate the safety of materials used in medical devices. We chose L929 cells because the cell line is recognized by the ISO 10993-5 standard and is widely accepted for cytotoxicity testing.

Our changes to the manuscript: The following statement was added to the SI:

“We chose L929 cells because the cell line is recognized by the ISO 10993-5 standard and is widely accepted for cytotoxicity testing.”

References:

- (1) Montalbano, G.; Toumpaniari, S.; Popov, A.; Duan, P.; Chen, J.; Dalgarno, K.; Scott, W. E.; Ferreira, A. M. Synthesis of bioinspired collagen/alginate/fibrin based hydrogels for soft tissue engineering. *Materials Science and Engineering: C* **2018**, *91*, 236-246. DOI: <https://doi.org/10.1016/j.msec.2018.04.101>.
- (2) Klippel, S.; Döpfert, J.; Jayapaul, J.; Kunth, M.; Rossella, F.; Schnurr, M.; Witte, C.; Freund, C.; Schröder, L. Cell Tracking with Caged Xenon: Using Cryptophanes as MRI Reporters upon Cellular Internalization. *Angew. Chem. Int. Ed.* **2014**, *53* (2), 493-496. DOI: <https://doi.org/10.1002/anie.201307290>.
- (3) Sahoo, D. R.; Biswal, T. Alginate and its application to tissue engineering. *SN Applied Sciences* **2021**, *3* (1), 30. DOI: 10.1007/s42452-020-04096-w.
- (4) Seiti, M.; Giuri, A.; Corcione, C. E.; Ferraris, E. Advancements in tailoring PEDOT: PSS properties for bioelectronic applications: A comprehensive review. *Biomaterials Advances* **2023**, *154*, 213655. DOI: <https://doi.org/10.1016/j.bioadv.2023.213655>.
- (5) Garbarczyk, J. E.; Wasiucionek, M.; Machowski, P.; Jakubowski, W. Transition from ionic to electronic conduction in silver–vanadate–phosphate glasses. *Solid State Ionics* **1999**, *119* (1), 9-14. DOI: [https://doi.org/10.1016/S0167-2738\(98\)00475-5](https://doi.org/10.1016/S0167-2738(98)00475-5).
- (6) Nguyen, D. M.; Wu, Y.; Nolin, A.; Lo, C. Y.; Guo, T.; Dhong, C.; Martin, D. C.; Kayser, L. V. Electronically Conductive Hydrogels by in Situ Polymerization of a Water-Soluble EDOT-Derived Monomer. *Advanced Engineering Materials* **2022**, *24* (10), 2200280-2200280. DOI: 10.1002/ADEM.202200280.
- (7) Nguyen, D. M.; Lo, C. Y.; Guo, T.; Choi, T.; Sundar, S.; Swain, Z.; Wu, Y.; Dhong, C.; Kayser, L. V. One Pot Photomediated Formation of Electrically Conductive Hydrogels. *ACS Polymers Au* **2024**, *4* (1), 34-44. DOI: 10.1021/ACSPOLYMERSAU.3C00031.

(8) Feig, V. R.; Tran, H.; Lee, M.; Bao, Z. Mechanically tunable conductive interpenetrating network hydrogels that mimic the elastic moduli of biological tissue. *Nat. Commun.* **2018**, *9* (1), 2740-2740. DOI: 10.1038/s41467-018-05222-4.

Response to Reviewers for Nature Communications manuscript NCOMMS-24-76582A

Reviewer #1 (Remarks to the Author):

The authors adequately reviewed the manuscript which improved significantly. I recommend its publication without further revisions.

In particular, all my comments have been addressed with additional experiments and explanation in the text when needed substantially improving clarity and strengthening their data to better support the conclusions.

With respect to reviewer 3, all the comments have been adequately addressed as well, including more clarification on the methods for assessing electrical/ electrochemical performance of the gels, conductivity calculations, elaboration on the structural changes at different concentrations, the gel healing time and finally cytocompatibility tests. Overall the authors have performed a very detailed revision with additional experiments, references to support the claims and more discussion in the text better clarifying the methodology, results and further supporting the conclusions.

Our response: Thank you for recognizing our efforts towards resolving your comments and those of reviewer 3, and for recommending our manuscript for publication.

Reviewer #2 (Remarks to the Author):

While the authors have made an effort to address the concerns raised in the initial round of reviews, several critical issues—particularly regarding data reliability and consistency across techniques—remain inadequately resolved. These deficiencies significantly compromise the validity of the structural and mechanistic claims made in the manuscript. I outline the major points below:

1. FFT and Cryo-EM Data Interpretation Remains Ambiguous:

a. The azimuthally integrated FFT data in Fig. S13g show negative "n" values and poorly resolved features. It is unclear how the spatial frequency data are derived or how peak assignments (e.g., 1.57–1.65 nm) are extracted.

b. The authors might incorrectly reference a “missing” 6 nm^{-1} signal, even though the maximum q in their FFT only reaches $\sim 3 \text{ nm}^{-1}$. The claimed absence of a signal cannot be used to conclude structural nonexistence without extending the q -range to at least 10 nm^{-1} .

Our response:

We thank the reviewer for their detailed review of the manuscript. Fig S13 and its caption indeed had incorrect labels, and units in panel h and lacked clarity likely because of a common discrepancy in how microscopy and scattering experts represent spatial frequencies and scattering

vectors. We have addressed these points and made changes to the manuscript, as summarized below.

Figure S13 shows that the PEDOT:PSS lamella visible in cryo-EM images cannot arise from artefacts of the microscope transfer function – an important detail for microscopy experts. Fig. S13g and Fig. S13h show radial integration of FFT figures in reciprocal space and line scans in real space of lamella features (seen in Fig. S13d-f), respectively. The latter real space line scan shows the peak-to-peak distance, highlighting the lamella spacing of PEDOT:PSS. We realize this may not have been clear because of the typo in the *x-axis* label of Fig. 13h; the axis label should have denoted d (nm) instead of q (nm^{-1}). We corrected the *x-axis* label and added more details on the caption of Fig. S13 to clarify what was represented. We also realize that the term “radial integration” is often clearer (and perhaps more correct) than “azimuthal integration”, and we now label Fig. S13g with “radial integration” in the revised caption for clarity. Thus, the negative d values in panel S13h are simply due to where we (arbitrarily) define zero in the real-space line scans.

In addition, the electron microscopy community traditionally presents the relationship between real space and reciprocal space vector using $d = 1/g$, which differs from how scattering results are usually represented, with $d = 2\pi/q$. As an example, 1.6 nm spacing corresponds to $g = 0.625 \text{ nm}^{-1}$ in FFTs extracted from cryo-EM micrograph, while the X-ray scattering peak corresponding to this spacing appears at $q = 3.925 \text{ nm}^{-1}$. We have now revised our manuscript to denote the reciprocal vector extracted from FFTs with g and the X-ray scattering reciprocal space vector as q . We have also modified the SI to highlight the difference between these two parameters for clarity. Thus, the missing signal denoted in the caption of Figure S13 near 0.6 1/nm is within the range of panel g.

Our changes to the manuscript:

We have modified Figure S13 by correcting any typos and clarifying results through edits to the caption. The modified caption and figure are included below.

Fig. S13. Cryo-EM micrographs of 1.8 wt% TR-CP at three different defocus values; large (3.5 μm), **a**, medium (1.9 μm), **b**, and small (1.5 μm) **c**, defocus values. EMAN2 software was used for contrast transfer function (CTF) fit and defocus calculation.¹ 0.14 nm pixel size, 0.01 mm spherical aberration, and 7% amplitude contrast values were used as initial input for the CTF fits. Panels **d**, **e**, and **f**, show the zoomed-in view of blue boxes shown in **a**, **b**, and **c**, respectively. White arrows denote lamella fringes observable in micrographs. **g**. Radial integration of FFTs of micrographs shown in panels **a**, **b**, and **c**, as the intensity versus spatial frequency/reciprocal space vector (g). The large number of oscillations is from CTF, which varies with defocus value. **h**. Line scans of the lattice fringes in real space as change in intensity versus distance. The peak-to-peak distance shows lamella spacing of PEDOT:PSS. Comparing **g** and **h**, the spacing of the lamella does not depend on defocus, indicating that the spacing corresponds to PEDOT:PSS structure and not an artefact from the oscillations in the CTF. As can be seen in panel **g**, radial integrated FFT profiles show no distinguishable peak associated with lamella spacing around 0.6 nm^{-1} (1.64 nm), aside from the CTF oscillations. Although FFTs of particular regions of interest, as shown in Fig. S14, show peaks associated with lamella spacing, FFTs of larger images do not exhibit these spacings. This is due to the low density of crystalline

domains in these micrographs, and is consistent with the lack of peaks associated with the lamella in solution SAXS results.

And have added the following citation:

(1) Tang, G.; Peng, L.; Baldwin, P. R.; Mann, D. S.; Jiang, W.; Rees, I.; Ludtke, S. J. EMAN2: An extensible image processing suite for electron microscopy. *Journal of Structural Biology* **2007**, *157* (1), 38-46. DOI: <https://doi.org/10.1016/j.jsb.2006.05.009>.

c. The connection between 6 nm^{-1} and 1.3 nm in the caption of Fig. S13 is not justified or physically explained.

Our response:

As it was stated in the previous answer, Fig. 13h shows the line scan in real space. We fixed the *x-axis* label of Fig. S13h, and better clarified the connection between the reciprocal space and real-space features.

d. In Fig. S13h, the real-space peak-to-peak distance decreases as q increases, which is contradictory to the authors' claim of invariant lamellar spacing. This trend implies dynamic changes in local structure or defocus artefacts, both of which need detailed clarification.

Our response:

The peak-to-peak distance in Fig. 13h shows the lamella spacing of PEDOT:PSS in real space. The similar peak-to-peak distance at different defocus values allows us to conclude that the lamella features are in fact a property of the samples and are not formed due to CTF in cryo-EM.

Our changes to the manuscript:

We modified the caption of Fig. S13 to address the reviewer's comment.

e. It remains unclear why the FFT-derived azimuthal profiles (panel g) are used to make statements about the absence of PEDOT stacking. The logic of assigning signal absence based on azimuthal—not radial—FFT profiles is questionable.

f. Furthermore, there is a notable inconsistency between panel g and panel h of Fig. S13. The data sources and processing parameters should be disclosed in more detail to explain this discrepancy.

Our response:

We believe that both these concerns are not resolved by fixing our labelling error. We also now clarify that Fig. S13g shows the radial integration of the FFT (integration over polar angle), while Fig. S13h shows the line scan in real space. Subsequently, the absence of a signal corresponding to lamella spacing is concluded from the 1D profile of radially integrated FFTs plots. We modified

the SI and Method section to state the use of DigitalMicrograph as the software for the generation and of FFT figures. We also stated that we use Fiji software for radial integration of 2D FFT graphs.

Our changes to the manuscript:

The following text has been added to the supplementary information (Section 1.7):

“Cryo-EM micrographs were collected at two different magnifications of 14000x and 47000x with a pixel size of 1 nm and 0.14 nm, respectively. Binning of 1 (no camera pixel binning) was used for the images with 4096×4096 pixels.

DigitalMicrograph software was used for the analysis of cryo-EM micrographs, including line scan and Fast Fourier Transform (FFT) analysis. Fiji software and radial integration plugin were used for radial integration of 2D FFT plots.² In FFT micrographs, the spatial frequency/reciprocal space vector (g) is inversely proportional to the real space distance ($d = 1/g$), as conventionally defined in microscopy analyses. On the other hand, the scattering vector in x-ray scattering results is related to real space distance as $d = 2\pi/q$. For example, a 1.6 nm spacing corresponds to $g = 0.625 \text{ nm}^{-1}$ in FFT graphs extracted from cryo-EM micrographs, while the X-ray scattering peak corresponding to this spacing appears at 3.925 nm^{-1} .”

And the following citation was added:

Schindelin, J.; Arganda-Carreras, I.; Frise, E.; Kaynig, V.; Longair, M.; Pietzsch, T.; Preibisch, S.; Rueden, C.; Saalfeld, S.; Schmid, B.; et al. Fiji: an open-source platform for biological-image analysis. *Nat. Methods* **2012**, *9* (7), 676-682. DOI: 10.1038/nmeth.2019.

2. SAXS Data Interpretation Is Weak and Inconsistent with Cryo-EM:
a. The SAXS data in Fig. R3 show clear peaks at $q = 5.6$ and 15 nm^{-1} . These peaks are only briefly addressed and ambiguously attributed to PNIPAM interactions or background, without proper experimental validation.

Our response: We have now included the original SAXS data for raw samples, empty holder, and water background, which are provided below in Fig. S16d, e. The data clearly shows that the peaks at $q = 15$ and 28 nm^{-1} were from the background and empty cell. As shown in the previous round of revisions, the peak at $q = 5.6 \text{ nm}^{-1}$ which is about 1.1 nm in real space is attributed to the intermolecular interaction from the PNIPAM as it is also present in the SAXS data for PSS-*b*-PNIPAM (no PEDOT), and appears only above the LCST (previously **Fig. R3**, now **Fig. R1** – this figure is soon going to be published in a separate unrelated manuscript on PSS-*b*-PNIPAM).

Fig. R1: Change in intensity as a function of q . Intensity profiles were recorded at different temperatures to capture the change in shape due to the LCST. Sample concentration = 10 mg/mL in DI water

Our changes to the manuscript:

We have modified Fig. S16 to the following:

Fig. S16. a. Core shell sphere model on Small-angle x-ray scattering results on TR-CP showing measurement, fitting lines and total fit b. below LCST and c. above LCST. d – e. Raw data from small-angle x-ray scattering for sample holder and water, compared to the TR-CP, d. below LCST and e. above LCST.

b. The authors' claim that the peaks at $q = 15$ and 28 nm^{-1} are background-related must be supported with separate background-subtracted plots or appropriate references. Without this, their exclusion from fitting is speculative and methodologically unsound.

Our response: The original SAXS data for raw samples, empty holder, and water background are provided below in comparison to the background subtracted SAXS data (and have been included in the manuscript, **Fig. S16 d, e**). The data clearly showed that the peaks at $q = 15$ and 28 nm^{-1} were from the background and empty cell. Background subtracted SAXS data (red) with the minimum background signal left after the background subtraction are due to composition mismatch and absorption mismatch from imperfect background matching of the SAXS system and sample cells – which cannot be prevented.

Fig. R2: Raw data from small-angle x-ray scattering for sample holder and water, compared to the TR-CP, **a.** below LCST and **b.** above LCST.

Our changes to the manuscript:

We have modified Fig. S16 to include the raw data from the background.

c. Discrepancy in the q -range for SAXS measurements at different temperatures (e.g., 20°C vs 40°C) raises serious questions about data comparability. The authors provide no explanation for this inconsistency.

Our response: We regret to say that we do not understand what the reviewer is referring to. There are no differences in the q range of SAXS data at different temperatures in Fig 3e and Fig S16. The raw SAXS data and background data have been added above for reference.

3. Cryo-EM Structural Claims Are Physically Incongruent:

a. The claim that lamellar fringes seen in Cryo-EM correspond to crystalline PEDOT is not reconciled with SAXS/WAXS data, where no corresponding features are found—even though PEDOT crystalline domains (~6 nm) should be visible in high-q scattering, as shown by prior literature (e.g., Chem. Mater. 2016, 28, 9, 3185–3192).

Our response:

We expect that the presence of a scattering peak corresponding to PEDOT depends on the number of crystals present. In our previous work, we also saw a lack of a peak corresponding to PEDOT:PSS lamella spacing in solution SAXS³, and this was consistent with other work.⁴ Thus, we attribute the lack of lamella features in our solution SAXS results to the low density of crystalline domains in the solution phase of our samples. On the other hand, as the reviewer points out, in the solid phase, lamella spacing features can be observed both in TEM and SAXS/WAXS results. In the solid phase, PEDOT:PSS shows a higher density of crystalline structures with larger crystalline domains⁵ compared to the solution phase. As a result, the scattering signal from these crystalline domains is stronger, which can be readily detected using SAXS/WAXS techniques.⁶

We were unable to locate the publication referenced by the reviewer. Therefore, we have supported our claims with other examples from literature.

Our changes to the manuscript:

We have added the following text to the SI:

“The lack of a peak corresponding to PEDOT:PSS lamella spacing in solution SAXS is consistent with prior reports.^{3,4} We attribute the lack of lamella features in solution SAXS to the low density of crystalline domains in the solution phase, low number of stacked chains, and overlap with background scattering. On the other hand, in the solid phase, PEDOT:PSS shows a higher density of crystalline structures with larger crystalline domains⁵ compared to the solution phase. As a result, the scattering signal from these crystalline domains is stronger, which can be readily detected using SAXS/WAXS techniques⁶.”

And have added the following references:

(3) Taussig, L.; Ghasemi, M.; Han, S.; Kwansa, A. L.; Li, R.; Keene, S. T.; Woodward, N.; Yingling, Y. G.; Malliaras, G. G.; Gomez, E. D.; et al. Electrostatic self-assembly yields a structurally stabilized PEDOT:PSS with efficient mixed transport and high-performance OECTs. *Matter* **2024**, 7 (3), 1071-1091. DOI: <https://doi.org/10.1016/j.matt.2023.12.021>.

(4) Takano, T.; Masunaga, H.; Fujiwara, A.; Okuzaki, H.; Sasaki, T. PEDOT Nanocrystal in Highly Conductive PEDOT:PSS Polymer Films. *Macromolecules* **2012**, *45* (9), 3859-3865. DOI: 10.1021/ma300120g.

(5) Gueye, M. N.; Carella, A.; Massonnet, N.; Yvenou, E.; Brenet, S.; Faure-Vincent, J.; Pouget, S.; Rieutord, F.; Okuno, H.; Benayad, A.; et al. Structure and Dopant Engineering in PEDOT Thin Films: Practical Tools for a Dramatic Conductivity Enhancement. *Chemistry of Materials* **2016**, *28* (10), 3462-3468. DOI: 10.1021/acs.chemmater.6b01035.

(6) Wang, Y.; Zhu, C.; Pfattner, R.; Yan, H.; Jin, L.; Chen, S.; Molina-Lopez, F.; Lissel, F.; Liu, J.; Rabiah, N. I.; et al. A highly stretchable, transparent, and conductive polymer. *Science Advances* **2017**, *3* (3), e1602076. DOI: doi:10.1126/sciadv.1602076.

b. The core-shell particle model (~5 nm radius) is inconsistent with the ~6 nm crystalline domains of PEDOT. If PEDOT were crystalline and formed the particle core, one would expect scattering or diffraction consistent with this structural scale.

Our response:

The crystalline PEDOT we observed by cryo-EM are lamellar fringes of 1.6 nm in spacing (in real space), it is therefore possible that structurally ~3 PEDOT chains would fit within the core of the spherical particles. Modifications to the manuscript are merged with those of the responses to the point below.

c. It is conceptually inconsistent that PEDOT is visible in Cryo-EM while the more abundant and chemically similar PSS and PNIPAM components are not. The authors must clarify the contrast mechanism, phase contrast vs Z-contrast, and consider charge density and hydration shell factors.

Our response:

We observe the combination of PEDOT, PSS, and PNIPAM (as micelles or “filomicelle” aggregates), but can see the PEDOT chains when they stack. Since we expect PSS and PNIPAM to be amorphous, we do not expect these chains to be identifiable beyond the micellar structures. We have clarified this point in the text.

Our changes to the manuscript:

We modified the main text and SI to explain the contrast mechanism (including a new section in the SI) and the reason behind the lack of observation of PSS and PNIPAM.

“Notes on Cryo-EM

The contrast observed in cryo-EM results arise from a combination of phase contrast due to differences in electrostatic (inner) potential between domains, and from differences in mass density, including between crystalline and amorphous regions.⁷ Subsequently, because PSS chains and PNIPAM are amorphous with similar chemical structures, contrast between PSS and PNIPAM domains is minimal. Previous reports show that PEDOT:PSS in films and dispersion phases consist of two different crystalline structures, namely, PEDOT:PSS core-shell micelles with a crystalline PEDOT core,⁴ and PEDOT:PSS lamella stacks with alternating PEDOT and PSS chains.³ In this study, domains containing the latter structures are referred to as crystalline domains, while the core-shell structures are referred to as core-shell micelles. We distinguish between these two different structures, as the core-shell micelles generally possess a more uniform size distribution,⁸ while the crystalline domains in conjugated polymers vary in size and shape within the same samples.⁹ Further studies are needed to elucidate the crystalline structure within the PEDOT-rich core of micelles.”

4. Possible overinterpretation and Misrepresentation of SAXS/WAXS Models:

a. The use of a core-shell sphere model oversimplifies the likely highly anisotropic, lamellar, or cylindrical features observed in Cryo-EM. The lamellar fringes (~20 nm long, ~5 nm wide) are inconsistent with spherical scattering models.

Our response:

We agree that the core-shell sphere model may not fully capture the heterogeneous structure of PEDOT:PSS in solution, but the reasonable fit of the scattering data to this model implies that core-shell micelles dominate the structure. Indeed, the low density of crystalline domains (lamellar fringes) in PEDOT:PSS solution would preclude a SAXS signature, and we propose that the scattering analyses provide a reasonable starting point for quantifying structural features. Given the limited features in the SAXS curve, the authors think that using more complex models may lead to overfitting and make the SAXS fitting results not robust. In addition, the core-shell model is used for modeling small spherical/ovoid particles, rather than for lamellar fringes. The effective radius has been used to express how close these particles are and reflect the particle assembling of the particles to flexible chains (below the LCST) or more tightly assembled gel networks (above the LCST).

Our changes to the manuscript:

We have edited the text in the manuscript to clarify this point, including the following text:

“We note that the SAXS did not capture the presence of the PEDOT lamella seen by cryo-EM due to their low concentration, limited number of stacked chains (i.e., small crystal size), and overlap with background scattering.” (bottom of page 10)

- b. The core radius of 3.2 nm (inferred from SAXS) contradicts literature reports (~30 nm) and the authors' own assertion that PEDOT forms ~6 nm crystals. This discrepancy is not addressed.
- c. The authors make unsubstantiated claims about lamellar features being “too large to be π - π stacking,” while lamellar π -stacking is routinely observed at similar dimensions in PEDOT-rich phases in aqueous dispersions via WAXS.

Our response:

The 3-5 nm core size calculated in this work is consistent with previous studies, which also report PEDOT-core in PEDOT:PSS with a similar size (a core radius of 2.4 ± 0.7 nm⁴). While at this time we cannot confirm the exact composition of the core, we can assume it contains PEDOT because of the high contrast and match with previous reports on PEDOT:PSS. If it contained only PEDOT, a core radius of 3.2 nm would imply that the PEDOT forms crystals with a size of ~6 nm, as the crystal size is double the radius. On the other hand, the 1.6 nm (not 6 nm) PEDOT crystalline domains, referenced here, are associated with the lamella structure of PEDOT:PSS. We further modified SI to clearly distinguish between core-shell micelles and PEDOT:PSS lamella structures.

We apologize for the confusion with regards to π - π stacking of PEDOT:PSS. We denote the lamella spacing to be about 1.36-1.5 nm, which is too large for pi stacking. Normally, π - π stacking is observed in a q of over 1.7 \AA^{-1} , and is already over the reliable q range of the SAXS measurement.^{10, 11} We have clarified our text to now better denote the value of the lamella spacing, such as in the caption of Figure S13.

Our changes to the manuscript:

We have edited the caption of Figure S13 to clarify this point (see above figure and caption).

Given the inconsistencies between techniques, questionable data interpretation, and unresolved structural ambiguities, I do not believe the manuscript in its current form meets the standards required for publication in Nature Communications. A complete revision with improved experimental methodology, transparent data processing, and consistent structure-property correlation is required before reconsideration.

Our response:

We believe that we have made appropriate changes to the manuscript and improved its clarity. We believe that much of the confusion came from mislabeling units and caption in Fig. S13. We also make the distinction between low concentration lamellar fringes of crystalline PEDOT and the bulk of the material which is spherical core-shell micelles, likely with a PEDOT-rich core (similar to accepted PEDOT:PSS models).

Additional references:

- (1) Tang, G.; Peng, L.; Baldwin, P. R.; Mann, D. S.; Jiang, W.; Rees, I.; Ludtke, S. J. EMAN2: An extensible image processing suite for electron microscopy. *Journal of Structural Biology* **2007**, *157* (1), 38-46. DOI: <https://doi.org/10.1016/j.jsb.2006.05.009>.
- (2) Schindelin, J.; Arganda-Carreras, I.; Frise, E.; Kaynig, V.; Longair, M.; Pietzsch, T.; Preibisch, S.; Rueden, C.; Saalfeld, S.; Schmid, B.; et al. Fiji: an open-source platform for biological-image analysis. *Nat. Methods* **2012**, *9* (7), 676-682. DOI: 10.1038/nmeth.2019.
- (3) Taussig, L.; Ghasemi, M.; Han, S.; Kwansa, A. L.; Li, R.; Keene, S. T.; Woodward, N.; Yingling, Y. G.; Malliaras, G. G.; Gomez, E. D.; et al. Electrostatic self-assembly yields a structurally stabilized PEDOT:PSS with efficient mixed transport and high-performance OECTs. *Matter* **2024**, *7* (3), 1071-1091. DOI: <https://doi.org/10.1016/j.matt.2023.12.021>.
- (4) Takano, T.; Masunaga, H.; Fujiwara, A.; Okuzaki, H.; Sasaki, T. PEDOT Nanocrystal in Highly Conductive PEDOT:PSS Polymer Films. *Macromolecules* **2012**, *45* (9), 3859-3865. DOI: 10.1021/ma300120g.
- (5) Gueye, M. N.; Carella, A.; Massonnet, N.; Yvenou, E.; Brenet, S.; Faure-Vincent, J.; Pouget, S.; Rieutord, F.; Okuno, H.; Benayad, A.; et al. Structure and Dopant Engineering in PEDOT Thin Films: Practical Tools for a Dramatic Conductivity Enhancement. *Chemistry of Materials* **2016**, *28* (10), 3462-3468. DOI: 10.1021/acs.chemmater.6b01035.
- (6) Wang, Y.; Zhu, C.; Pfattner, R.; Yan, H.; Jin, L.; Chen, S.; Molina-Lopez, F.; Lissel, F.; Liu, J.; Rabiah, N. I.; et al. A highly stretchable, transparent, and conductive polymer. *Science Advances* **2017**, *3* (3), e1602076. DOI: doi:10.1126/sciadv.1602076.

- (7) Newcomb, C. J.; Moyer, T. J.; Lee, S. S.; Stupp, S. I. Advances in cryogenic transmission electron microscopy for the characterization of dynamic self-assembling nanostructures. *Current Opinion in Colloid & Interface Science* **2012**, *17* (6), 350-359. DOI: <https://doi.org/10.1016/j.cocis.2012.09.004>.
- (8) Murphy, R. J.; Weigandt, K. M.; Uhrig, D.; Alsayed, A.; Badre, C.; Hough, L.; Muthukumar, M. Scattering Studies on Poly(3,4-ethylenedioxythiophene)–Polystyrenesulfonate in the Presence of Ionic Liquids. *Macromolecules* **2015**, *48* (24), 8989-8997. DOI: 10.1021/acs.macromol.5b02320.
- (9) Pokuri, B. S. S.; Stimes, J.; O'Hara, K.; Chabinyk, M. L.; Ganapathysubramanian, B. GRATE: A framework and software for GRaph based Analysis of Transmission Electron Microscopy images of polymer films. *Computational Materials Science* **2019**, *163*, 1-10. DOI: <https://doi.org/10.1016/j.commatsci.2019.02.030>.
- (10) Rivnay, J.; Inal, S.; Collins, B. A.; Sessolo, M.; Stavrinidou, E.; Strakosas, X.; Tassone, C.; Delongchamp, D. M.; Malliaras, G. G. Structural control of mixed ionic and electronic transport in conducting polymers. *Nat. Commun.* **2016**, *7* (1), 11287. DOI: 10.1038/ncomms11287.
- (11) Paulsen, B. D.; Wu, R.; Takacs, C. J.; Steinrück, H.-G.; Strzalka, J.; Zhang, Q.; Toney, M. F.; Rivnay, J. Time-Resolved Structural Kinetics of an Organic Mixed Ionic–Electronic Conductor. *Advanced Materials* **2020**, *32* (40), 2003404. DOI: <https://doi.org/10.1002/adma.202003404>.

Response to Reviewers for Nature Communications manuscript NCOMMS-24-76582A

Reviewer #4 (Remarks to the Author):

This work explores a thermo-responsive conducting polymer that undergoes a fully reversible non-covalent crosslinking at 35 °C within less than a minute to form conductive hydrogels. The thermo-responsive conducting polymer is based on a PEDOT:PSS-b-PNIPAM block copolymer. A material with outstanding performance.

The manuscript in the revised form has been significantly improved; however, there are still several issues that need to be clarified before its publication:

Our response: Thank you for reviewing our manuscript and for providing great remarks.

1. In the caption of Figure S12, the meaning of values 53 and 57 nm should be explained.

Our response: We have modified the caption to explain what the 53 and 57 nm values mean: the particle-to-particle distance and mesh size, respectively.

Our changes to the manuscript: New caption for Fig. S12 (now, Fig. S17 due to editorial changes)

Fig S17. Cryo-EM micrographs and their Fast Fourier Transforms (FFTs), shown as inset, for TR-CP. **a.** 1.8 wt%, below LCST. **b.** 3.6 wt%, below LCST **c.** 1.8 wt%, above LCST showing the formation of spherical structures spaced by 53 nm. **d.** 3.6 wt%, above LCST showing the formation of a bi-continuous network with a mesh size of 57 nm.

2. In the caption of Figure S13, should be indicated if it is either below or above LCST.

Our response: We have edited the caption of Fig S13 (now, Fig. S17 due to editorial changes) to include “above LCST”.

Our changes to the manuscript: Edited caption for Fig. S13 (now, Fig. S18 due to editorial changes)

Fig. S18. Cryo-EM micrographs of 1.8 wt% TR-CP, above LCST, at three different defocus values; **a.** large (3.5 μm), **b.** medium (1.9 μm), and **c.** small (1.5 μm) defocus values. EMAN2⁵ software was used for contrast transfer function (CTF) fit and defocus calculation. 0.14 nm pixel size, 0.01 mm spherical aberration, and 7% amplitude contrast values were used as initial input for the CTF fits. Insets: 2D FFTs of the corresponding images. Panels **d**, **e**, and **f**, show the zoomed-in view of blue boxes shown in **a**, **b**, and **c**, respectively. White arrows denote lamella fringes observable in micrographs. **g.** Azimuthal integration of FFTs of micrographs shown in panels **a**, **b**, and **c**, as the intensity versus spatial frequency/reciprocal space vector (**g**). The large number of oscillations is

from CTF, which varies with defocus value. Azimuthal integration was carried out using Fiji software with the built-in Radial Integration plugin. **h**. Line scans of the lattice fringes in real space as change in intensity versus distance. The peak-to-peak distance shows lamella spacing of PEDOT:PSS. Comparing **g** and **h**, the spacing of the lamella does not depend on defocus, indicating that the spacing corresponds to PEDOT:PSS structure and not an artefact from the oscillations in the CTF. As can be seen in panel **g**, Azimuthal integrated FFT profiles show no distinguishable peak associated with lamella spacing around 0.6 nm^{-1} (1.64 nm), aside from the CTF oscillations. Although FFTs of particular regions of interest, as shown in Fig. S14, show peaks associated with lamella spacing, FFTs of larger images do not exhibit these spacings. This is due to the low density of crystalline domains in these micrographs, and is consistent with the lack of peaks associated with the lamella in solution SAXS results.

3. The FFTs from micrographs a), b), and c) in Figure S13 should be included, and indicated how the azimuthal integration has been performed.

Our response: The FFTs have been included in Fig S13 (now, Fig S18). The caption has been edited to include how the azimuthal integration has been performed.

Our changes to the manuscript: Edited figure and caption for Fig. S18 as shown below.

Fig. S18. Cryo-EM micrographs of 1.8 wt% TR-CP, above LCST, at three different defocus values; **a.** large (3.5 μm), **b.** medium (1.9 μm), and **c.** small (1.5 μm) defocus values. EMAN2⁵ software was used for contrast transfer function (CTF) fit and defocus calculation. 0.14 nm pixel size, 0.01 mm spherical aberration, and 7% amplitude contrast values were used as initial input for the CTF fits. **Insets: 2D FFTs of the corresponding images.** Panels **d,** **e,** and **f,** show the zoomed-in view of blue boxes shown in **a,** **b,** and **c,** respectively. White arrows denote lamella fringes observable in micrographs. **g.** Azimuthal integration of FFTs of micrographs shown in panels **a,** **b,** and **c,** as the intensity versus spatial frequency/reciprocal space vector (**g**). The large number of oscillations is from CTF, which varies with defocus value. **Azimuthal integration was carried out using Fiji software with the built-in Radial Integration plugin.** **h.** Line scans of the lattice fringes in real space as change in intensity versus distance. The peak-to-peak distance shows lamella spacing of PEDOT:PSS. Comparing **g** and **h,** the spacing of the lamella does not depend on defocus, indicating that the spacing corresponds to PEDOT:PSS structure and not an artefact from the oscillations in the CTF. As can be seen in panel **g,** Azimuthal integrated FFT profiles show no distinguishable peak associated with lamella spacing around 0.6 nm^{-1} (1.64 nm), aside from the CTF oscillations. Although FFTs of particular regions of interest, as shown in Fig. S14, show peaks associated with lamella spacing, FFTs of larger images do not exhibit these spacings. This is due to the low density of crystalline domains in these micrographs, and is consistent with the lack of peaks associated with the lamella in solution SAXS results.

3. The FFTs from micrographs a), b), and c) in Figure S13 should be included, and indicated how the azimuthal integration has been performed.

4. The term “azimuthal integration” instead of “radial integration” is extensively used for the X-ray scattering community in order to derive the $I(q)$ profiles from 2D patterns.

Our response: We have changed the nomenclature to “azimuthal integration”.

Our changes to the manuscript: Edited caption for Fig. S

Fig. S18. Cryo-EM micrographs of 1.8 wt% TR-CP, above LCST, at three different defocus values; **a.** large (3.5 μm), **b.** medium (1.9 μm), and **c.** small (1.5 μm) defocus values. EMAN2⁵ software was used for contrast transfer function (CTF) fit and defocus calculation. 0.14 nm pixel size, 0.01 mm spherical aberration, and 7% amplitude contrast values were used as initial input for the CTF fits. **Insets: 2D FFTs of the corresponding images.** Panels **d,** **e,** and **f,** show the zoomed-in view of blue boxes shown in **a,** **b,** and **c,** respectively. White arrows denote lamella fringes observable in micrographs. **g.** **Azimuthal** integration of FFTs of micrographs shown in panels **a,** **b,** and **c,** as the intensity versus spatial frequency/reciprocal space vector (**g**). The large number of oscillations is from CTF, which varies with defocus value. **Azimuthal** integration was carried out using Fiji software with the built-in Radial Integration plugin. **h.** Line scans of the lattice fringes in real space as change in intensity versus distance. The peak-to-peak distance shows lamella spacing of PEDOT:PSS. Comparing **g** and **h,** the spacing of the lamella does not

depend on defocus, indicating that the spacing corresponds to PEDOT:PSS structure and not an artefact from the oscillations in the CTF. As can be seen in panel **g**, **Azimuthal** integrated FFT profiles show no distinguishable peak associated with lamella spacing around 0.6 nm^{-1} (1.64 nm), aside from the CTF oscillations. Although FFTs of particular regions of interest, as shown in Fig. S14, show peaks associated with lamella spacing, FFTs of larger images do not exhibit these spacings. This is due to the low density of crystalline domains in these micrographs, and is consistent with the lack of peaks associated with the lamella in solution SAXS results.

5. Concerning Figure S16d) and e), what is the background subtracted to the raw data? The authors could try to subtract the signals from the sample holder and water.

Our response: What we called “background” was the signals from the holder and the water which were subtracted from the sample in what we called “TR-CP, background subtracted”.

Our changes to the manuscript: We have clarified in the caption of Fig. S16 (now, fig. S21)

Fig. S21. a. Core shell sphere model on Small-angle x-ray scattering results on TR-CP showing measurement, fitting lines and total fit **b.** below LCST and **c.** above LCST. **d – e.** Raw data from small-angle x-ray scattering for sample holder and water, compared to the TR-CP, **d.** below LCST and **e.** above LCST. ‘Background subtracted’ means that the signals from the sample holder and water were subtracted from the TR-CP measurements.

6. What is the meaning of the Gaussian Peak 1 (centered at about 0.1 nm^{-1}) in Figure S16c)?

Our response: This peak corresponds to the length scale of 61 nm, which represents the large aggregates formed above LCST.

Our changes to the manuscript: We have added the following sentence to Supplementary Information:

“In this work, the SAXS of TR-CP sample above the LCST exhibits a scattering peak centered at 0.102 nm^{-1} (marked as Gaussian Peak 1 in **Fig. S21C**), which corresponds to a characteristic length scale of 61 nm.”

The following sentence is added to the main text:

“A strong structure factor peak $S(q)$ appeared in low q range (centered at 0.102 nm^{-1}). These features indicate that after gelation, the particles aggregated to form large, ordered structures with a center-to-center distance of $\sim 60 \text{ nm}$, in line with the cryo-EM results at $45 \text{ }^\circ\text{C}$.”